# Integrated Power and Propulsion System Optimization for a Planetary-Hopping Robot

Himangshu Kalita, Alvaro Diaz-Flores and Jekan Thangavelautham *

Space and Terrestrial Robotic Exploration (SpaceTREx) Laboratory, Aerospace and Mechanical Engineering Department, University of Arizona, Tucson, AZ 85721, USA
* Correspondence: jekan@arizona.edu

**Abstract:** Missions targeting the extreme and rugged environments on the moon and Mars have rich potential for a high science return, although several risks exist in performing these exploration missions. The current generation of robots is unable to access these high-priority targets. We propose using teams of small hopping and rolling robots called SphereX that are several kilograms in mass and can be carried by a large rover or lander and tactically deployed for exploring these extreme environments. Considering that the importance of minimizing the mass and volume of these robot platforms translates into significant mission-cost savings, we focus on the optimization of an integrated power and propulsion system for SphereX. Hydrogen is used as fuel for its high energy, and it is stored in the form of lithium hydride and oxygen in the form of lithium perchlorate. The system design undergoes optimization using Genetic Algorithms integrated with gradient-based search techniques to find optimal solutions for a mission. Our power and propulsion system, as we show in this paper, is enabling, because the robots can travel long distances to perform science exploration by accessing targets not possible with conventional systems. Our work includes finding the optimal mass and volume of SphereX, such that it can meet end-to-end mission requirements.

**Keywords:** design; planetary exploration; power and propulsion; robotics





## 1. Introduction

Exploring the far corners of the solar system by performing orbital, surface, and subsurface exploration with the help of human and robotic explorers will define the next wave of space exploration. These exploration platforms will reach diverse surface environments on the moon and Mars, including caves, canyons, cliffs, skylights, and craters. They will answer fundamental questions about the origins of the solar system, origins of life, and prospects for In situ Resource Utilization (ISRU). These are some of the high-priority targets, as outlined in [1]. Some of these environments, such as caves and lava tubes, are well insulated from radiation, harsh weathering such as dust storms, and high and low external temperatures. These insulated environments could harbor isolated, ancient ecosystems or remains of the last survivors. They can also provide insight as a potential habitat for future human explorers. Current robotic missions are unable to reach these places of interest, mainly due to precision landing limitations, an inability to reach or move in rugged environments, and the risks involved in adverse mission culture. This is despite the great potential science that is possible from taking these exploratory risks. Hence, there is an important need for novel robotic systems that can access and explore these rugged environments [2]. A credible solution is to develop mother–daughter architectures that permit daughter crafts to take high exploratory mission risks for high reward science but without putting the mothercraft or the rest of the mission in jeopardy. One such example is the Mars helicopter, Ingenuity, that hitched a ride to Mars on the Perseverance rover and demonstrated technologies to test powered, controlled flight on another world for the first time [3].

Here we present a small, modular, low-cost, spherical robot platform called SphereX that is designed to explore rugged environments such as caves, lava tubes, pits, and canyons on the moon, Mars, icy moons, comets, and asteroids. SphereX operates by hopping and rolling short distances [4,5]. SphereX is designed with space-grade electronics, including a command and data-handling board, power management/distribution board, and radio with S/X-band antennas for communication/coordination among teams of robots and with a mothercraft. Mobility is enabled using a single thruster-propulsion system combined with a 3-axis reaction wheel-based attitude control to perform ballistic hops. Rolling is enabled using the attitude control-system alone. The onboard power supply contains a power generation, storage, and distribution subsystems. The remaining volume contains a stereo camera and a LiDAR system for 3D imaging, surveying, and navigation. A flagship rover or lander can carry several SphereX robots that can be deployed on demand to access and explore extreme environments that would otherwise be inaccessible to the rover or lander. Here, we propose an integrated power supply and propulsion system for SphereX consisting of Proton-exchange membrane (PEM) fuel cells for power and a $H_2/O_2$ propulsion system as shown in Figure 1. Hydrogen is generated on demand through the reaction of water and lithium hydride (LiH) [6]. Oxygen is generated on demand through a catalytic decomposition of lithium perchlorate ($LiClO_4$) [7]. The hydrogen fuel and oxidizer are transported to the fuel cell system using micropumps while pressurized nitrogen gas is used to transport the reactant to the propulsion combustion chamber. Wastewater from the PEM fuel cell is collected and recycled to generate hydrogen on demand using the lithium hydride (LiH) hydrogen generator. Based on lessons learned from our earlier research [8], we intend to implement a fuel cell/battery hybrid system to maximize mission life. Here, our work focuses on finding the optimal mass and volume of SphereX that meet the constant power demand for a predefined mission life while exploring a target distance. There is an important need to optimize the mass and volume of SphereX due to the high launch cost involved in integrating planetary exploration robots, instruments, and platforms.

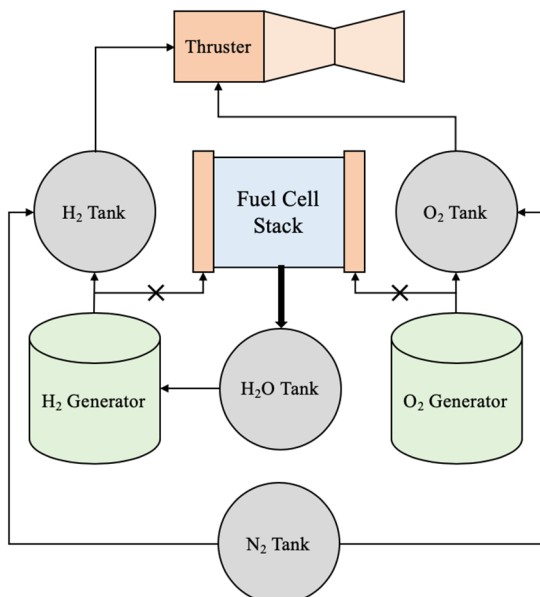

**Figure 1.** Integrated power supply and propulsion system for the SphereX robot platform.

Finding the optimal design of SphereX in terms of volume and mass (which in turn impacts cost and mission life) is a highly coupled problem between multiple disciplines.

Traditionally, space systems are optimized through the rigorous evaluation of each discipline independently. Although feasibility is achieved with this labor-intensive approach, there is no guarantee for achieving an optimal overall system. Thus, the design of SphereX

could benefit from the application of Multidisciplinary Design Optimization (MDO). However, complexity arises in an MDO approach to design SphereX due to challenges in modeling, the use of structurally complex and discontinuous objective cost function and design space, and a wide range of constraints. Here, we approach this problem by applying a hybrid optimization method by applying Genetic Algorithm (GA)-based optimization integrated with multiple gradient-based optimization sequences [9]. The methodology used in this research can find near optimal designs in terms of mass, volume, and assembly feasibility for SphereX applied to different missions. Before going into the next sections, we first present our main contributions in this paper.

- We propose an integrated power supply and propulsion system for SphereX consisting of Proton-exchange membrane (PEM) fuel cells for power and a $H_2/O_2$ propulsion system.
- We develop a system-level optimization problem to find optimal design solutions of SphereX in terms of mass, volume, and power.
- A comparative analysis of the optimal design solution of SphereX is provided against other relevant systems.

In the following sections, we present background information on fuel cells, miniaturized propulsion systems, and multidisciplinary optimization approaches used in the past (Section 2). This is followed by Section 3, where we present the details of all the subsystems of SphereX, along with the formulation of the system-level optimization problem and details of the optimization algorithm used. Section 4 presents the results found using the proposed algorithm for two different mission scenarios, which is followed by a comparative analysis between the proposed system and other similar systems in Section 5. Section 6 presents the conclusion of the paper.

## 2. Background

Fuel cells have been suggested for a multitude of autonomous robotic systems and sensor network modules such as for powering hopping robots [10], ground robots [11], humanoid robots [12], unmanned underwater vehicles (UUVs) [13], and landers and rovers for moon and Mars exploration [14,15]. Fuel cells being clean, quiet, and offering high efficiency and specific energy makes them an ideal option as a power source for robotic systems to be deployed in pristine environments such as caves and lava tubes on the surface of the moon and Mars. However, factors such as the degradation of fuel-cell life due to fluctuations in power demand and the efficient storage of hydrogen and oxygen prevents it from becoming a widespread energy source [8]. One proposed application demonstrates a hybrid system where a battery is recharged with the fuel cell, which is used to power the system [8,16]. For robot systems such as SphereX, which is small in mass and volume, the cryogenic storage of hydrogen and oxygen at high pressures is not practical. For such applications, storing hydrogen and oxygen in solid form is viable and has been considered before. One method is to release hydrogen from solid hydride through depressurization; however, it remains unattractive, due to low storage efficiencies in the order of 0.5% to 2.5% [17]. Hydrides that use heat to release hydrogen can reach up to 18% storage efficiency, but substantial energy and infrastructure are required to reach temperatures of 70 to 800 °C [17]. Another approach is to produce hydrogen through the hydrolysis reaction of metal hydrides with water. Similarly, oxygen can be stored in the form of metal oxides. The Soyuz spacecraft used chemical oxygen generators with potassium superoxide, while the International Space Station (ISS) and Mir used a Vika oxygen generator with lithium perchlorate [18].

Miniaturized propulsion technology for space systems has relied heavily on low-specific-impulse and low-thrust technologies. However, the propulsion system for SphereX would require producing a thrust greater than its weight in order to perform ballistic hopping on the surface of the moon and Mars. Cold gas or pulsed plasma systems are fairly well established and demonstrated for small delta-v missions, but the specific impulse and thrust developed are fairly low. A liquid sulfur hexafluoride cold gas thruster has

been used in a 3U CubeSat mission Can X-2 and attained a specific impulse of 50 s [19]. The propulsion system for the MarCo Mission uses a R-236fa refrigerant with a specific impulse of 40 s [20]. Electrospray propulsion systems have not only achieved significant maturity, but also have the disadvantage of low thrust in the order of 1mN [21]. Solid rocket motors are a good option for SphereX, due to their overall simplicity, technology maturity, and high specific impulse and thrust. However, for performing multiple hops with SphereX, solid rockets are not ideal, as they expend all their propellant at once in addition to producing vibrations and have difficulty in throttling. A compelling alternative is to use Polymer Electrolyte Membranes to electrolyze water into hydrogen and oxygen on-demand and to produce high thrust and specific impulse through combustion. An electrolysis propulsion system was designed that offers a specific impulse of 400 s, but the power required is between 50 to 200 W [22]. Another electrolysis propulsion system called Hydros was developed that can deliver up to 0.25–0.6 N of thrust at 258 s of specific impulse [20]. However, major challenges in electrolysis propulsion systems remain in preventing the water from freezing and a high-power requirement.

With an overview of fuel cells and a miniaturized propulsion system described above, designing them together for a robotic system to meet mission requirements is a complex task. Aerospace system designers have been using multidisciplinary design optimization (MDO) approaches to a variety of aerospace problems for decades now. However, typically, most of the efforts have been focused on designing aircraft structures and space launch vehicles. Its application to miniaturized space systems such as planetary exploration robots is minimal. The launch vehicle design completed by Olds et al. [23] is the first identified MDO application in space systems. MDO was first applied to design satellites by Matossian as demonstrated when designing an Earth-observing satellite mission [24]. Later, the application of MDO to develop space-system engineering tools was continued by Riddle [25], Bearden [26], George et al. [27], Fukunaga et al. [28], Mosher et al. [29], Stump et al. [30], and Barnhart et al. [31]. Ravanbakhsh et al. [32] worked on structural design-sizing tool using an MDO approach. An evaluation of optimization techniques was provided by Taylor et al. [33] applied to complex spacecraft design problems. A multi-objective, multidisciplinary design optimization method for space systems was developed by Jilla et al. [34]. The application of the distributed Collaborative Optimization (CO) method was shown by Jafarsalehi et al. [35] for small-satellite missions. The method used gradient-based techniques at the discipline level and gradient-free GA at the system level. The MDO literature provided multiple instances where system engineering tools were developed using mathematical optimization techniques to design complex systems comprising multiple disciplines.

Motivated by these ideas, our approach includes designing a combined power and propulsion system for SphereX by storing hydrogen and oxygen in solid form. Moreover, we use an MDO approach to optimize the overall design of SphereX considering predefined mission requirements.

## 3. System Design

This section provides a detailed description of each subsystem of SphereX along with the design specification used. First, we provide a detailed description of the fuel cell subsystem, followed by the propulsion subsystem, and then the hydrogen and oxygen generators, followed by the storage tanks and other relevant subsystems. With all the subsystems described, we then present the formulation of the system-level optimization problem along with details of the genetic algorithm used to find optimal solutions for our problem.

### 3.1. PEM Fuel Cells

The working principle of PEM fuel cells used as a power source is discussed in detail in Appendix A. The goal of this optimization effort is to find the optimal operating voltage $V$ and current density $i$ of the fuel cell system, such that the number of cells $n$ and the

mass of oxygen $m_{O_2,fc}$ and hydrogen $m_{H_2,fc}$ used is minimized, satisfying a constant power demand $P$ for a mission lifetime $\Gamma$. The design variables are $\mathbb{x}_{fc} = [n, i]$. The values of $n$, $m_{O_2,fc}$, and $m_{H_2,fc}$ are normalized between $[0, 1]$, represented by $\underline{n}$, $\underline{m}_{O_2,fc}$ and $\underline{m}_{H_2,fc}$ based on their upper and lower bounds, and the objective is a weighted sum of the normalized values. The power generated $P_{fc}$ by the fuel cell system is the product of the total voltage of the fuel cell stack $V_{fc} = nV$ and the current $I_{fc} = nai$, where $a$ is the MEA surface area of each cell. One constraint is added to the optimization problem, such that the power generated $P_{fc}$ is greater than or equal to the power demand $P$. The optimization problem is mathematically described as:

$$min f_{fc}\left(\mathbb{x}_{fc}\right) = \alpha_1 \underline{n} + \alpha_2 \underline{m}_{O_2,fc} + \alpha_3 \underline{m}_{H_2,fc} \tag{1a}$$

$$subject\ to\ \left\{ g_{fc}\left(\mathbb{x}_{fc}\right) \equiv P_{fc} \geq P \right. \tag{1b}$$

The optimization problem defined here is a mixed-integer optimization problem, as the design variable $n$ can take only integer values. Thus, we use genetic algorithms to solve the optimization problem. Figure 2 shows the value of the objective function over the number of generations with the weights $\alpha_1 = \alpha_2 = \alpha_3 = 1$, MEA surface area $a = 10\ cm^2$, and power demand $P = 16\ W$. The optimal values of the design variables are the number of cells $n = 3$, the current density $i = 194.38\ mA/cm^2$, and the operating voltage $V = 0.9146\ V$.

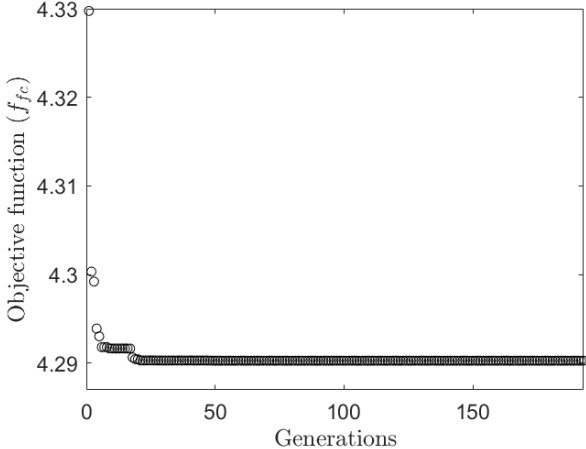

**Figure 2.** Objective function value with number of generations for the optimization problem.

### 3.2. Propulsion System

The dynamics of the robot while using the combined action of the propulsion system and the attitude-control system for performing ballistic hops is presented in Appendix B. To perform that, the critical subsystem for SphereX is the propulsion system. Each SphereX contains one primary lift engine at the bottom that uses $H_2$ fuel and $O_2$ oxidizer. Given that SphereX is limited in mass and volume, the propulsion system uses nitrogen gas to transport the reactants into the combustion chamber, hence avoiding the use of pumps and mechanical devices. The thermodynamic parameters of the combustion products, namely combustion temperature $T_c$, molar mass $M$, and ratio of specific heats $\gamma$, were determined using the Cpropep software for a given combustion pressure $p_c$ and mixture ratio $r_m = m_{O_2,\ prop}/m_{H_2,\ prop}$ [36]. Figure 3 shows the variation of combustion temperature, molar mass, and ratio of specific heats against the combustion pressure, mixture ratio, and the corresponding specific impulse for an exit pressure of 200 Pa. An experiment was performed to collect data by using the Cpropep software, and a model was fitted using a radial basis function. The thermodynamic model is then used for optimizing the propulsion subsystem.

**Figure 3.** Variation of (**a**) combustion temperature $T_c$(K); (**b**) molar mass $M$ (g/mole); (**c**) ratio of specific heats $\gamma$; and (**d**) specific impulse $I_{sp}$ (s).

For the robot (SphereX) with mass $m$ and radius $r$ to perform ballistic hopping on the surface of a planet with gravity $g$, a propulsion system is required to generate thrust, such that $||F|| > mg$. For our analysis, we have considered $||F|| = 2mg$, and we need to design each element of the propulsion system. Figure 4 shows the basic elements of cylindrical combustion chamber and conical nozzle. For the propulsion system to produce a constant thrust of $||F||$, the throat area of the nozzle is:

$$A_{th} = \frac{||F||}{p_c\sqrt{\frac{2\gamma^2}{\gamma-1}\left(\frac{2}{\gamma+1}\right)^{\frac{\gamma+1}{\gamma-1}}\left[1-\left(\frac{p_e}{p_c}\right)^{\frac{\gamma-1}{\gamma}}\right]}+(p_e-p_a)\epsilon} \tag{2}$$

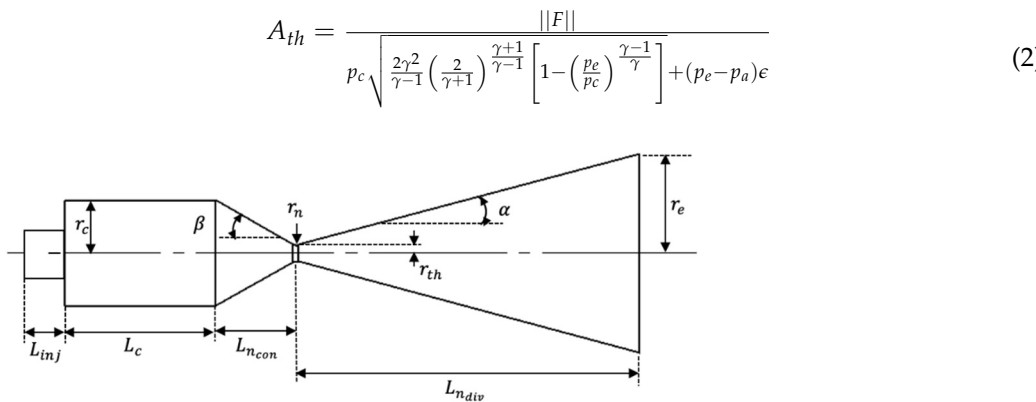

**Figure 4.** Design elements of basic cylindrical combustion chamber and conical nozzle.

The theoretical nozzle expansion ratio $\epsilon$ is then expressed as:

$$\epsilon = \frac{A_e}{A_{th}} = \frac{\mathfrak{V}}{\sqrt{\frac{2\gamma}{\gamma-1}\left(\frac{p_e}{p_c}\right)^{\frac{2}{\gamma}}\left[1-\left(\frac{p_e}{p_c}\right)^{\frac{(\gamma-1)}{\gamma}}\right]}} \tag{3}$$

where, $\mathfrak{V} = \sqrt{\gamma(2/(\gamma+1))^{(\gamma+1)/(\gamma-1)}}$ is the Vandenkerckhove function [37]. For a conical nozzle, the nozzle throat section is defined by its radius, convergent cone half, and divergent cone half angle. The radius $r_n$ ranges from 0.5 to 1.5 times the throat radius, the convergent cone half angle $\beta$ ranges from 20° to 45°, and the divergent cone half angle $\alpha$ ranges from 12° to 18°. The length of the diverging and converging conical nozzle section can then be expressed as:

$$L_{n(div)} = \frac{r_{th}(\sqrt{\epsilon}-1)+r_n(\sec\alpha-1)}{\tan\alpha}; \; L_{n(con)} = \frac{r_{th}(\sqrt{\epsilon_c}-1)+r_n(\sec\beta-1)}{\tan\beta}, \tag{4}$$

where, $r_{th}$ is the radius of the nozzle throat, and $\epsilon_c = A_c/A_{th}$ is the contraction area ratio (where, $A_c$ is the area of the combustion chamber). For a propellant with characteristic length $L^*$, the time needed for vaporization and the complete reaction of the propellants, known as residence time, is $\tau^* = L^*c^*/RT_c$. The minimum chamber volume $\mathcal{V}_c$ required can then be determined as, $\mathcal{V}_c = \tau^*\dot{m}RT_c/p_c$, where, $\dot{m} = A_{th}p_c\sqrt{\gamma(2/(\gamma+1))^{(\gamma+1)/(\gamma-1)}/RT_c}$ is the mass flow rate. Finally, the length of the combustion chamber can be calculated as $L_c = \mathcal{V}_c/A_c$. Moreover, the length of the injector $L_{inj}$ is estimated to be 25% of the sum of the length of the nozzle and the combustion chamber. As such the total length is defined as $L_{total} = L_{n(div)} + L_{n(con)} + L_c + L_{inj}$. Considering the combustion chamber to be a cylindrical shell, the wall thickness is, $t_c = FSp_cr_c/\sigma$, where $r_c$ is the radius of the combustion chamber, $\sigma$ is the tensile strength of the material used, and $FS$ is a safety factor. The thickness of the converging and diverging section of the nozzle is assumed to be the same as the combustion chamber wall thickness. The mass of the combustion chamber and nozzle $m_{c+n}$ is then calculated by multiplying the total surface area $S_{c+n}$ with the wall thickness $t_c$ and density of material used $\rho$ as $m_{c+n} = S_{c+n}t_c\rho$. Moreover, the mass of injectors, sensors, control equipments, valves, fittings, pipes, and other necessary interfaces are estimated as 20% of the propulsion system's dry mass $m_{dry}$.

Next, we calculate the mass of propellant required for the robot to explore a target distance $d_{targ}$. The delta-v, $\Delta v_{hop}$, required for the robot to perform a single hop of distance $d_{hop}$ is calculated by solving the optimal control problem for ballistic hopping as discussed in Section 4. Hence, the robot needs to perform $n = d_{targ}/d_{hop}$ hops, and, as such, the total delta-v required to explore the target distance is $\Delta v_{total} = n\Delta v_{hop}$. The mass of propellant required is then calculated as $m_{prop} = m\left(1 - e^{-\Delta v_{total}/I_{sp}g_0}\right)$, where the specific impulse $I_{sp}$ is

$$I_{sp} = \frac{1}{g_0}\left(\sqrt{2RT_c\left(\frac{\gamma}{\gamma-1}\right)\left[1-\left(\frac{p_e}{p_c}\right)^{\frac{\gamma-1}{\gamma}}\right]} + \frac{(p_e-p_a)A_e}{\dot{m}}\right). \tag{5}$$

Thus, the mass of the oxidizer and fuel is calculated as $m_{O_2,prop} = r_m m_{prop}/(r_m+1)$ and $m_{H_2,prop} = m_{prop}/(r_m+1)$.

Optimization

The objective of the optimization process for the propulsion subsystem is to minimize the total mass of the propulsion system, $m_{p(total)} = m_{dry} + m_{prop}$. The design variables are combustion pressure $p_c$, exit pressure $p_e$, mixture ratio $r_m$, and contraction area ratio $\epsilon_c$, $\mathbb{x}_p = [p_c, p_e, r_m, \epsilon_c]$. One constraint is added to the optimization function such that the total length $L_{total}$ is less than 90% of the radius of the robot. The mass $m$ and radius $r$ of the

robot, target exploration distance $d_{targ}$ and single hopping distance $d_{hop}$ should be given as the input. The optimization problem is the following:

$$min f_p(\mathbb{x}_p) = m_{p(total)} \tag{6a}$$

$$subject \ to: \ g_p(\mathbb{x}_p) \equiv L_{total} < 0.9r \tag{6b}$$

The design problem is modeled as a nonlinear optimization problem (NLP), and a sequential quadratic programming (SQP) method is used to solve it. At each iteration of SQP, the gradients and the hessian of the objective function and the constraints need to be calculated. It is performed using the finite difference method and the Broyden–Fletcher–Goldfarb–Shanno (BFGS) method respectively. The variations of the objective function value, constraint violation, and the first-order optimality of the optimization are shown in Figure 5. The objective function value remains stationary, and the constraint violation and the first-order optimality approach zero.

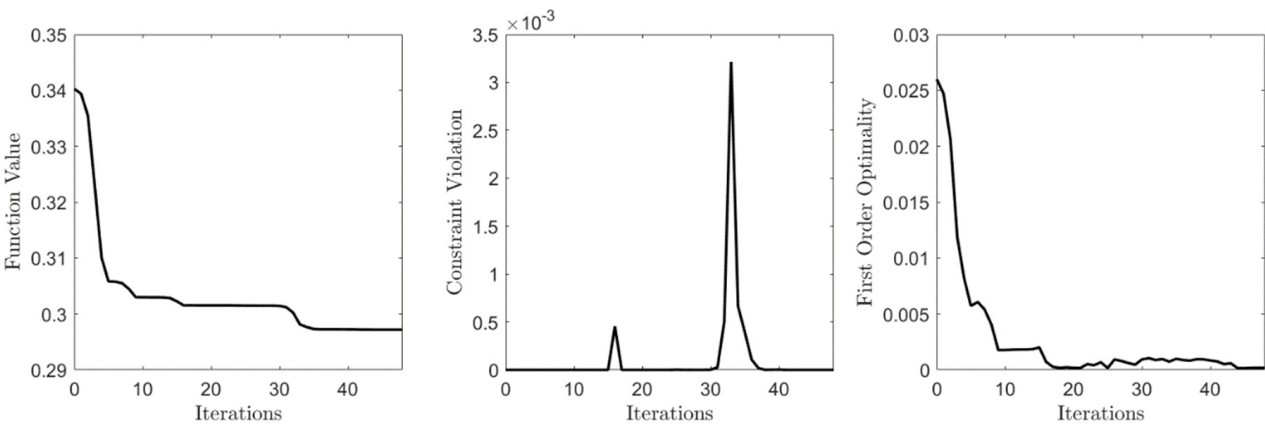

**Figure 5.** Variations of the objective function, constraint violation, and first-order optimality over iterations for the propulsion subsystem showing convergence of the optimization problem.

### 3.3. Hydrogen Generator

One of the major factors preventing $H_2$ from becoming a widespread energy source is the challenges involved in efficient storage. The cryogenic storage of $H_2$ at high pressure is not practical for small, low-power robotic systems such as SphereX because of complexity and difficulty in integrating them in such a small volume. Metal hydrides are considered for SphereX because of easy storage capabilities and their high gravimetric hydrogen weight densities. Metal hydrides react with water through the process of hydrolysis to release hydrogen and produce a metal hydroxide, some of which are shown in Table 1.

**Table 1.** Gravimetric hydrogen content of various metal hydrides.

| Hydride | Hydrogen Weight Content |
|---|---|
| $LiBH_4$ | 18.4% |
| $NaBH_4$ | 10.6% |
| $Be(BH_2)_4$ | 20.8% |
| $NaAlH_4$ | 7.4% |
| $NaH$ | 4.1% |
| $CaH_2$ | 4.7% |
| $LiH$ | 12.5% |

Among the metal hydrides considered, the hydrogen production rates for sodium borohydride and lithium borohydride are slow and require expensive catalysts for reaction completion. The hydrogen content of beryllium borohydride is very high, but it is too reactive and toxic. Alkali metal hydrides such as lithium hydride, sodium hydride, and

calcium hydride can produce hydrogen on contact with water and do not require a catalyst. LiH has the highest hydrogen weight content and is selected for SphereX. LiH is composed of lithium metal bonded with hydrogen and reacts with liquid water or water vapor to produce $H_2$ gas and LiOH according to the following reaction [38,39]:

$$LiH + H_2O \rightarrow LiOH + H_2$$

Experiments have shown that the above reaction can achieve 95–100% reaction-completion rates with an excess of water [38,39]. The molar mass of hydrogen is $2.014 \times 10^{-3}$ kg/mole, and that of lithium hydride is $7.95 \times 10^{-3}$ kg/mole. Thus, the amount of lithium hydride required is:

$$m_{LiH} = 3.9435 \left( m_{H_2,fc} + m_{H_2,prop} \right) \tag{7}$$

Similarly, for each mole of $H_2$ produced, one mole of $H_2O$ is required. The PEM fuel cell will produce water at a rate of $\dot{H_2O} = 9.34 \times 10^{-8} P/V$ kg/s, which will be reused for generating $H_2$ from LiH. The molar mass of water is $18.015 \times 10^{-3}$ kg/mole, thus the amount of water required is:

$$m_{H_2O} = 8.9373 \left( m_{H_2,fc} + m_{H_2,prop} \right) - \dot{H_2O}\Gamma, \tag{8}$$

where, $\Gamma$ is the mission lifetime.

### 3.4. Oxygen Generator

In the absence of atmospheric oxygen for operating the fuel cells, oxygen must be stored for both the fuel cells and propulsion system. Similar to hydrogen storage, the high pressure and cryogenic storage of oxygen is also not practical for small, low-power applications such as SphereX. However, chemical oxygen generators can be used that store metal oxides. When combined with a catalyst under sufficient temperature, the resulting chemical reaction releases oxygen and produces heat. Table 2 shows some examples of metal oxides as a source of oxygen.

**Table 2.** Gravimetric oxygen content of various metal oxides.

| Oxide | Oxygen Weight Content |
|---|---|
| $NaClO_3$ | 45.1% |
| $BaO_2$ | 18.9% |
| $KClO_4$ | 46.2% |
| $KO_2$ | 45.0% |
| $LiClO_4$ | 60.1% |
| $BeCl_2O_8$ | 61.5% |

Oxygen candles produce oxygen by igniting a mixture of sodium chlorate and iron powder at about 600 °C. Oxygen generators used in commercial airlines contain a mixture of sodium chlorate, 5% barium peroxide, and 1% potassium perchlorate, which releases oxygen once ignited at 260 °C. Soyuz spacecraft used chemical oxygen generators with potassium superoxide, where $KO_2$ reacts with $H_2O$ and $CO_2$ to produce oxygen. Another proposed solution to produce oxygen is to use tetramethylammonium ozonide because of its low molecular weight and 39% oxygen content. In terms of oxygen weight content, lithium perchlorate has one of the highest and has already been used in the Vika oxygen generator on Mir and the International Space Station (ISS). Although beryllium perchlorate has the highest oxygen weight content, it is expensive and toxic. The thermal decomposition of lithium perchlorate at 400 °C results in a 60% by-weight release of oxygen according to the following reactions:

Main reaction: $LiClO_4 \rightarrow LiCl + 2O_2$; side reaction: $4LiClO_4 \rightarrow 2Li_2O + 7O_2 + 2Cl_2$

The fast decomposition of pure $LiClO_4$ as seen in the main reaction requires a temperature of 400 °C. However, the high temperature leads to the weak development of the side reaction. Hence, some catalysts are added in order to increase the $LiClO_4$ decomposition rate. Transition-metal oxides such as $Fe_2O_3$, $Co_2O_3$, $TiO_2$, $MnO_2$, and $CuO$ increase the decomposition rate of perchlorates. Moreover, $Li_2O_2$ and $Li_2O$ have been studied for the catalyzed decomposition of $LiClO_4$ [40]. From the main reaction of decomposition of $LiClO_4$ for two moles of oxygen, one mole of lithium perchlorate is required. The molar mass of oxygen is $32 \times 10^{-3}$ kg/mole, and that of lithium perchlorate is $106.39 \times 10^{-3}$ kg/mole. Thus, the amount of lithium perchlorate required is:

$$m_{LiClO_4} = 1.6624 \left( m_{O_2,fc} + m_{O_2,prop} \right) \tag{9}$$

Moreover, energy is required for the decomposition reaction of $LiClO_4$, and the energy consumed $E_h$ for heating $LiClO_4$ from ambient temperature $T_a$ to 400 °C is calculated as follows:

$$E_h = m_{LiClO_4} c_p (400 - T_a), \tag{10}$$

where $c_p$ is the heat capacity of $LiClO_4$.

### 3.5. Storage Tanks

Storage tanks need to be designed for the hydrogen generator to store LiH and $H_2O$, and the oxygen generator to store $LiClO_4$. However, lithium hydride, when converted to lithium hydroxide, expands to three times its original volume. Moreover, the hydrogen and oxygen generated from the hydrogen and oxygen generator, respectively, need to be stored in separate tanks before being delivered to the combustion chamber of the propulsion system. The amount of hydrogen and oxygen stored in the hydrogen and oxygen tanks is such that the robot can perform at least five hops with it. The pressure drop across the injector orifice and solenoid valve is estimated to be approximately 1 MPa, hence the hydrogen and oxygen are stored at a pressure $p_{H_2/O_2\ tank} = p_c + 10^6$ pa. The hydrogen and oxygen will be transported to the fuel-cell stack through micro pumps directly from the hydrogen and oxygen generators [41]. However, for the propulsion system, the design uses pressurized nitrogen gas to initiate the transport of hydrogen and oxygen into the combustion chamber. The nitrogen is stored at pressure $p_{N_2\ tank} = 2p_{H_2/O_2\ tank}$. The mass of nitrogen required for transport of hydrogen and oxygen to the combustion chamber is:

$$m_{N_2} = V_{H_2\ tank}\ \rho_{N_2} \left( p_{H_2\ tank} \right) + V_{O_2\ tank}\ \rho_{N_2} \left( p_{O_2\ tank} \right), \tag{11}$$

where $V_{(*)\ tank}$ is the volume of the tank storing compound $(*)$, and $\rho_{(*)} \left( p_{(\sim)} \right)$ is the density of compound $(*)$ at pressure $p_{(\sim)}$. Considering the tanks to be a spherical shell, the volume and wall thickness of all the tanks are determined from the mass and density:

$$V_{(*)tank} = \alpha \frac{m_{(*)}}{\rho_{(*)} \left( p_{(\sim)tank} \right)};\ t_{(*)tank} = FS \frac{p_{(*)tank} r_{(*)tank}}{2\sigma}, \tag{12}$$

where $\alpha = 3$ for LiH tank, and $\alpha = 1$; otherwise, $\sigma$ is the tensile strength of the material used, $r_{(*)\ tank}$ is the radius of the tank storing compound $(*)$, and $FS$ is a safety factor. Subsequently, the mass of each tank is:

$$m_{(*)tank} = 4\pi r_{(*)tank}^2\ t_{(*)tank}\ \rho, \tag{13}$$

where $\rho$ is the density of material used. Moreover, it should be noted that the mass of hydrogen and oxygen used for calculating the tank dimensions is as follows: $m_{H_2/O_2} = m_{H_2/O_2,fc} + m_{H_2/O_2,prop}$.

### 3.6. Other Subsystems

In addition to the power-generation and propulsion subsystems, SphereX will accommodate other subsystems such as attitude control, command and data handling, communication, instruments, power management, and external shell. The details of the attitude control, command and data handling, communication, and instruments used are provided in Appendix B. This section provides a brief description of the power management and shell subsystem.

### 3.6.1. Power Management

For SphereX to explore an unknown environment, a Hop+Map→Stop→Process cycle is employed, with each cycle taking 180 s. SphereX hops from its current position to a desired position, simultaneously performing the mapping of the environment with the onboard LiDAR and stereo cameras during its Hop+Map phase. After the Hop+Map phase is over, the Stop phase is initiated, where the robot stops, and then the Process phase is initiated, where the robot processes the collected data, and then the subsequent Hop+Map phase is triggered. As the robot operates in a series of different phases, each subsystem of the robot operates at different cycles, resulting in a varying power demand, as shown in Figure 6a. The life of the fuel cell is significantly reduced when connected to a varying load, due to voltage oscillations. As such, the power system is designed as a fuel cell/battery hybrid system, where the battery is constantly charged by the fuel cell, while the battery along with a power-management board handles the varying demands of the load as shown in Figure 6b.

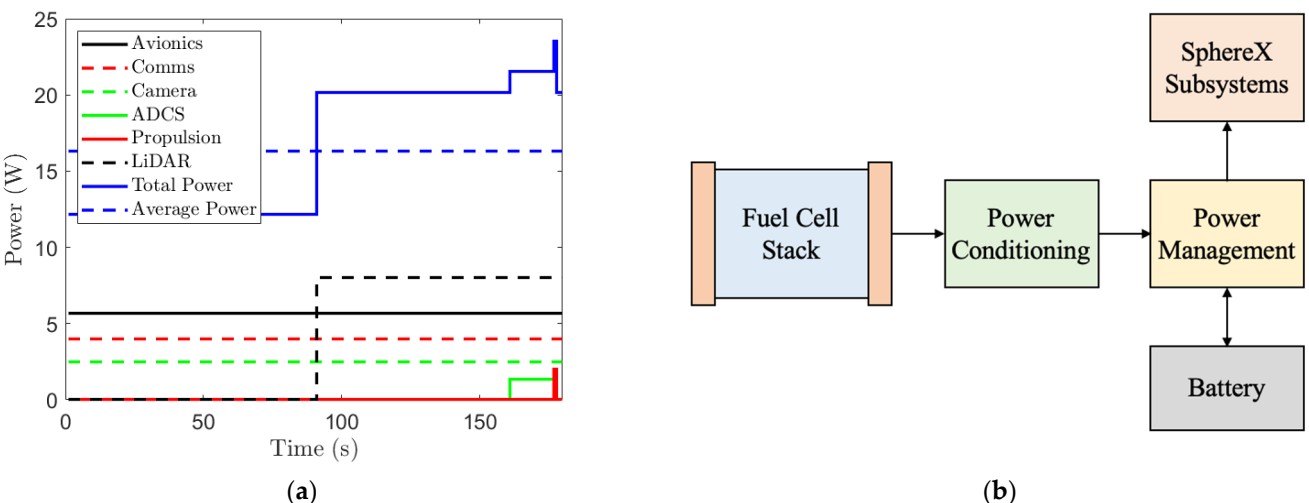

**Figure 6.** (**a**) Estimated power consumption profile by each subsystem of SphereX during a cycle of 180 s. (**b**) Proposed fuel cell power supply system.

The battery selected for our design is the GomSpace NanoPower Lithium Ion 18,650 which has a capacity of 2600 mAh @ 3.7 V, mass of 48 g, and dimensions of length 65 mm and radius 18.5 mm. With lithium-ion batteries selected for power storage, the GomSpace NanoPower P31u board is selected for power management, which is compatible with the batteries. The board offers the maximum power-point tracking (MPPT) capability and measures and logs currents, voltages, and temperatures of the power system. The board consumes only 0.165 W power under nominal conditions, weighs only 100 g, and has a dimension of $89.3 \times 92.9 \times 15.3$ mm. However, the number of batteries required needs to be optimized. The State of Charge ($SOC$) of the battery is $\dot{SOC} = P_{bat}/V_{bat}Q$, where $Q$ is the nominal discharge capacity of the battery, $P_{bat}$ is the input power at any given time instant to the battery, and $V_{bat}$ is the voltage output of the battery. We can compute the power at any given time-instant as the sum of all loads as $P_{bat} = P_{fc} - P_{prop} - P_{ac} - P_{comm} - P_0$,

where $P_{fc}$ is the power generated by the fuel cell system, $P_{prop}$ is the power consumed by the propulsion system, $P_{ac}$ is the power consumed by the attitude-control system, $P_{comm}$ is the power consumed by communication subsystem, and $P_0$ is the power consumed by the avionics board and scientific instruments. The battery voltage output based on *SOC* is:

$$V_{bat}(SOC) = \left(3 + \frac{e^{SOC} - 1}{e - 1}\right)\left(2 - e^{\lambda\frac{T - T_0}{T_0}}\right) \tag{14}$$

In order to avoid voltage oscillations of the fuel cell, the battery used for power storage should not experience a substantial voltage drain when connected to the load. As such, the selection of the battery is very crucial; it should have enough capacity to maintain the voltage level. The design specification is such that the battery does not operate at a voltage less than 3.5 V, which corresponds to an 85.5% state of charge. Hence, the objective of the optimization process is to minimize the discharge capacity of the battery system, such that the state of charge *SOC* during the entire mission is greater than 85.5%, which is expressed as (18). The design variable is $\mathbb{x}_b = Q$.

$$\min f_b(\mathbb{x}_b) = Q \tag{15a}$$

$$subject\ to:\ g_b(\mathbb{x}_b) \equiv SOC(\Gamma) > 0.855 \tag{15b}$$

This subsystem is also modeled as a nonlinear optimization problem (NLP), and a sequential quadratic programming (SQP) method is used to solve it. Based on the optimum discharge capacity, we calculate the number of batteries required to be $n_{bat} = ceil(Q/2.6)$ and mass to be $m_{bat} = 48n_{bat}$. The batteries are stacked in columns of 4 cells inside a cuboid of size $l = 0.065$, $b = 0.0185\min(n_{bat}, 4)$, and $h = 0.0185ceil(n_{bat}/4)$.

3.6.2. Shell

This discipline determines the optimal thickness of the SphereX shell for its structural rigidity upon impacting a surface with velocity $v$. An empirical approximation obtained using Zener [42] and Boettcher, Russell, and Mueller [43] measures the deformation $\delta$ (deflection of impact surface of sphere toward center of sphere) of a hollow sphere to obtain the following:

$$\delta = 0.67\left(\frac{\rho v^2}{\sigma}\right)^{\frac{1}{2}}\left(\frac{r}{t}\right)^{0.08} r, \tag{16}$$

where, $r$ is the radius, $t$ is the thickness of the sphere, $\rho$ is the density, and $\sigma$ is the ultimate stress of the sphere's material. The objective function is to find the minimum thickness of the shell, such that the deformation is less than $\delta_{\max}$, which is a user-defined value. The design variable is the shell thickness, $\mathbb{d}_s = t$ and the optimization problem is described as:

$$\min f_s(\mathbb{x}_s) = t \tag{17a}$$

$$subject\ to:\ g_s(\mathbb{x}_s) \equiv \delta < \delta_{max} \tag{17b}$$

This subsystem is also modeled as a nonlinear optimization problem (NLP), and a sequential quadratic programming (SQP) method is used to solve it. With the optimal thickness determined, the mass of the shell is determined to be $m_s = 4\pi r^2 t\rho$.

*3.7. System Optimization*

The objective of the system optimization process is to find the optimal mass $m$ and radius $r$ of the robot (SphereX) that accommodate all the subsystems while satisfying mission-specific requirements: (a) power demand, (b) mission lifetime, and (c) target-exploration distance. The task is formulated as a single-objective optimization problem with two design variables $\mathbb{x} = [m, r]$ and two constraints. Based on the values of mass

$m$ and radius $r$ with bounds $m^{(b)} = \begin{bmatrix} m^{(L)} & m^{(U)} \end{bmatrix}$ and $r^{(b)} = \begin{bmatrix} r^{(L)} & r^{(U)} \end{bmatrix}$, it is normalized between $[0 \ 1]$ as shown:

$$\underline{m} = \frac{m - m^{(L)}}{m^{(U)} - m^{(L)}}, \underline{r} = \frac{r - r^{(L)}}{r^{(U)} - r^{(L)}} \tag{18}$$

The objective function is then defined as $f(\mathbb{x}) = \alpha_1 \underline{m} + \alpha_2 \underline{r}$. The first constraint is defined, such that the difference between the design variable $m$ and the total mass of all the subsystems $m_{total}$ is equal to zero. The second constraint is that the assembly index is equal to one, the details of which are presented in Appendix C. The optimization process can then be formulated as:

$$\min f(\mathbb{x}) = \alpha_1 \underline{m} + \alpha_2 \underline{r} \tag{19a}$$

$$subject \ to: \ g_1(\mathbb{x}) \equiv m - m_{total} = 0 \tag{19b}$$

$$g_2(\mathbb{x}) \equiv Index = 1 \tag{19c}$$

This problem is posed as a constrained optimization problem, as such the search space is divided into two regions: feasible and infeasible regions. The optimal solution found must also lie in the feasible region. As such, the nonlinear optimization problem with two nonlinear equality constraints stated above is solved using the Augmented Lagrangian Genetic Algorithm (ALGA) [44]. However, the bounds of the design variables $\mathbb{x}$ are handled separately. To solve this optimization problem, the objective function and the two nonlinear constraint functions are combined using Lagrangian and penalty parameters to formulate a subproblem in the form of a cost function $\mathcal{J}(\mathbb{x}, \lambda, \mathcal{P})$ as shown in Equation (20).

$$\mathcal{J}(\mathbb{x}, \lambda, \mathcal{P}) = f(\mathbb{x}) + \sum_{i=0}^{2} \lambda_i g_i(\mathbb{x}) + \frac{\mathcal{P}}{2} \sum_{i=0}^{2} (g_i(\mathbb{x}))^2 \tag{20}$$

where the components $\lambda_i$ of the vector $\lambda$ are nonnegative and are known as the Lagrange multiplier estimates, and $\mathcal{P}$ is a positive penalty parameter.

With each subproblem solution representing a generation, a genetic algorithm is used to minimize the cost function, while making sure that the bounds are satisfied. The value of the Lagrangian multipliers and the penalty parameter are initialized at each generation, and the cost function is minimized. While minimizing each subproblem, the values of $\lambda$ and $\mathcal{P}$ are kept fixed for that particular generation. In the subsequent generations, the estimates of the Lagrangian multipliers $\lambda$ are updated, such that the subproblem is minimized within a required accuracy while satisfying the constraints. If the constraints are not met, the penalty parameter is increased by a penalty factor, which results in a new subproblem formulation, and the genetic algorithm minimizes the problem. These steps are iterated until the stopping criteria are met.

With the cost function for the constrained-optimization problem defined, the working principle of a genetic algorithm-based optimization technique to find the optimal values of the design variables is described here, as shown in Figure 7a. The algorithm starts by creating a random parent population $P_0$ of size $N_P$. Each individual of the population comprises the values of $m$ and $r$ chosen with a uniform distribution $m = \mathcal{U}(m^{(U)}, m^{(L)})$ and $r = \mathcal{U}(r^{(U)}, r^{(L)})$. Next, the value of the objective function $f(\mathbb{x})$ is evaluated using the subsystem discipline models as shown in Figure 7b for each individual. With the initial estimates of the Lagrange multipliers $\lambda$ and the penalty parameter $\mathcal{P}$, the fitness value is calculated for each individual according to the cost function $\mathcal{J}(\mathbb{x}, \lambda, \mathcal{P})$. After the fitness values are calculated, each individual participates in the evolution process through the genetic operators such as: selection, crossover, and mutation. This creates an offspring population $Q_0$ of size $N_Q$ [45]. For the selection operator, the tournament selection method is used. For the crossover operator, a blend crossover (BLX-$\alpha$) method is used [46]. For the mutation operator, a non-uniform mutation method is used [47]. In each generation, elitism is introduced by comparing the current population with previously found best solutions. For the $t^{th}$ generation, a combined population $R_t = P_t + Q_t$ is formed of size $N + N_Q$. Next,

the fitness of each individual in $R_t$ is calculated according to the cost function $\mathcal{J}(\mathbb{x}, \lambda, \mathcal{P})$. Based on the fitness value, the population is sorted in ascending order, and the best $N$ individuals are selected for the next generation. The new population $P_{t+1}$ of size $N$ is again used for selection, crossover, and mutation to create a new population $Q_{t+1}$ of size $N_Q$. The process is repeated until the desired results are obtained or the max number of generations is achieved.

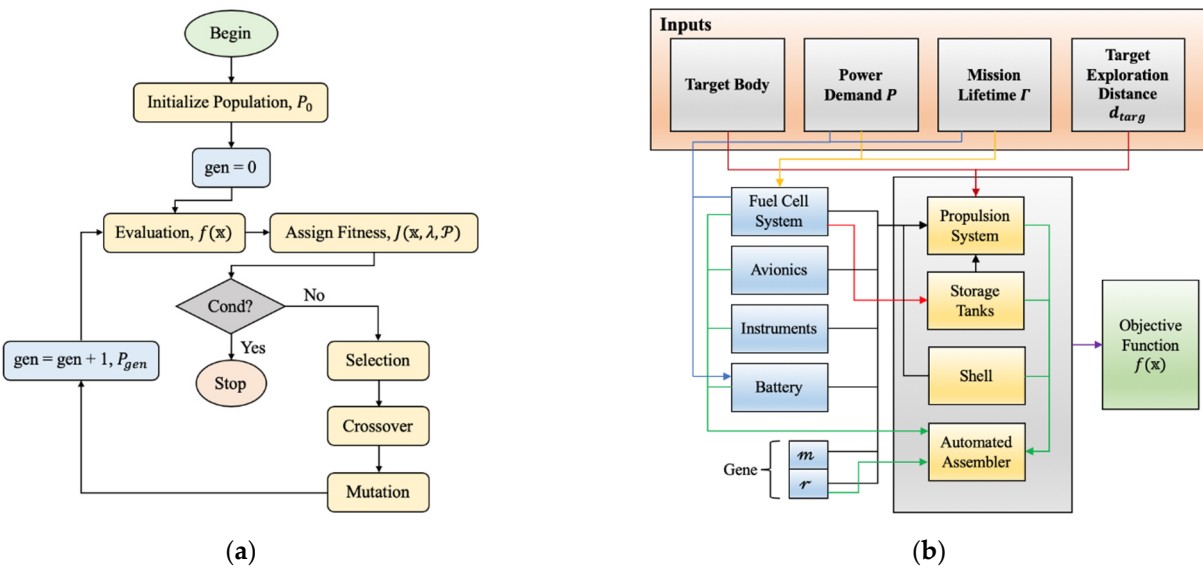

(**a**)  (**b**)

**Figure 7.** (**a**) Schematic of Genetic Algorithm. (**b**) Calculation of cost function value from subsystem models of SphereX.

## 4. Results

For our simulation, the genes are coded with real values of $n$ and $n$ with the number of populations, $N_P = 50$. Table 3 shows the materials used for the design of the nozzle, combustion chamber, storage tanks, and robot shell. Carbon fiber is the material selected for the robot shell. Given its high tensile strength and the low gravity on the lunar surface, we expect minimal challenges due to shock and vibration. Furthermore, the onboard thrusters can be used to nullify or minimize the experienced impact force to minimize risks.

**Table 3.** Material used for the design of the nozzle, combustion chamber, storage tanks, and robot shell.

| Component | Material | Tensile Strength $\sigma$ (MPa) | Density $\rho$ (kg/m$^3$) |
| --- | --- | --- | --- |
| Nozzle + Combustion chamber | Stainless Steel | 215 | 7700 |
| Storage Tanks | Aluminum | 324 | 2780 |
| Shell | Carbon fiber | 3500 | 2000 |

### 4.1. Test Scenario 1: Subsurface Exploration of Mare Tranquilitatis Pit

The first simulation was run to perform the sub-surface exploration of Mare Tranquilitatis Pit on the surface of the moon at 8.33° N 33.22° E. The 10 m/pixel-resolution images from the Terrain Camera (TC) aboard the Japanese lunar orbiter Selenological and Engineering Explorer (SELENE) and 1 m/pixel-resolution images from the high-resolution Narrow-Angle Camera (NAC) onboard the Lunar Reconnaissance Orbiter Camera (LROC) revealed the dimensions of the pit. The long axis of the pit is measured to be 98 m, the short axis to be 84 m, and the depth measured from shadow measurements to be 107 m. Moreover, there is evidence that the pit opens into a sub-lunarean void of at least 20 m in extent. With the possibility of kilometers of lava-tube extension below this pit, the mission specification is defined as exploring at least 1000 m of the void. The concept of operations

for performing such an exploration mission is shown in Figure 8a. A lunar lander carrying multiple SphereX robots would land near Mare Tranquilitatis Pit and deploy the robots. Each robot will undergo three phases of operation starting with 1. surface maneuvers to approach the pit; 2. pit entrance maneuver; and 3. sub-surface operation to explore the pit. Our goal is to explore 500 m on the surface in 2.5 h. In addition, it would take 10 min to enter the 50 m pit and travel 1000 m inside the lava tube in 5 h, as seen in Figure 8b.

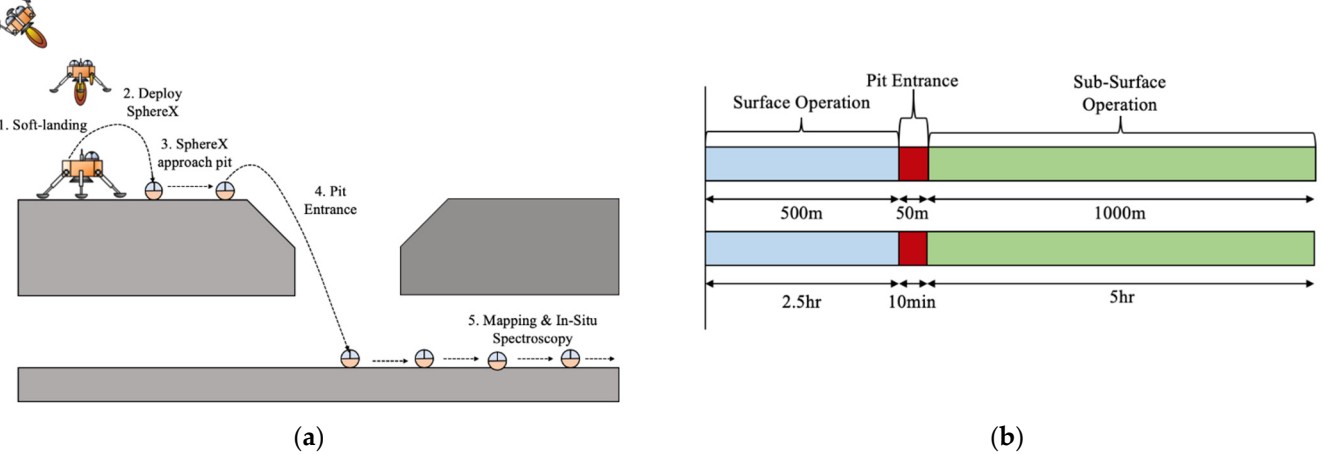

(a)                                                                                     (b)

**Figure 8.** (**a**) Concepts of operation for exploring Lunar pits. (**b**) Mission exploration requirements for the robot to enter the pit and explore.

For entering the pit, the robot needs a soft-landing maneuver, as such, in addition to the three phases discussed for the hopping controller in Appendix A, the robot has an additional soft-landing phase as shown in Figure 9a. For the soft-landing phase, the control angle $\gamma$ can be derived as $\gamma = \tan^{-1}(v_z/v_x)$, as shown in Figure 9b. The dynamics of the robot are:

$$\dot{r} = v, \dot{v} = \begin{cases} g + \frac{F}{m} \\ g \\ g + \frac{F}{m} \end{cases}, \dot{m} = \begin{cases} -\frac{||F||}{I_{sp}g_0} & if\ t < t_b \\ 0 & if\ t_b < t < t_l \\ -\frac{||F||}{I_{sp}g_0} & if\ t_l < t < \tau \end{cases} \tag{21}$$

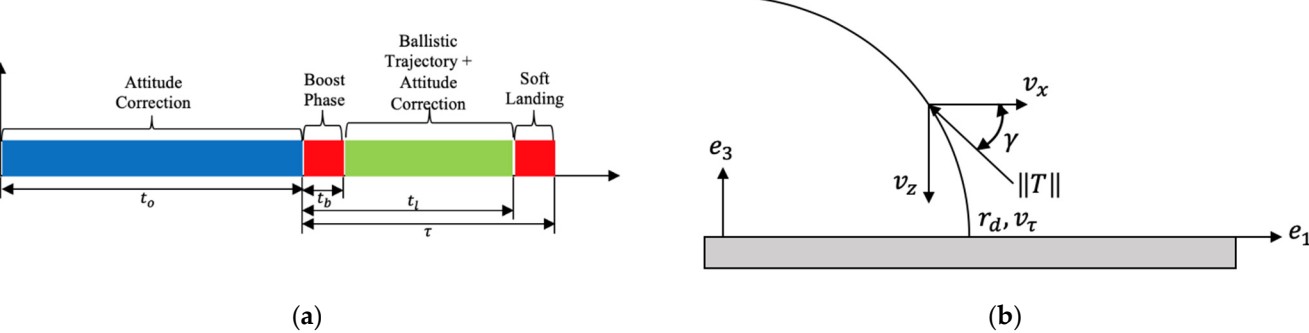

(a)                                                                                     (b)

**Figure 9.** (**a**) Time diagram for soft-landing ballistic hopping to enter Lunar pits. (**b**) Description of the control-angle $\gamma$ for soft-landing.

The objective of the optimization process is to minimize the fuel consumption and the optimal index can be expressed as $f(\mathbb{x}_{sl}) = \int_0^{t_b} ||F||dt + \int_{t_l}^{\tau} ||F||dt$. Four constraints are added, with the design variables $\mathbb{x}_{sl} = [t_b, t_l, \tau]$, and the optimization problem is as follows:

$$\min f_{sl}(\mathbb{x}_{sl}) = \int_0^{t_b} ||F||dt + \int_{t_l}^{\tau} ||F||dt \tag{22a}$$

$$subject\ to:\ g_{sl1}(\mathbb{x}_{sl}) \equiv t_b - t_l < 0 \tag{22b}$$

$$g_{sl2}(\mathbb{x}_{sl}) \equiv t_l - \tau < 0 \tag{22c}$$

$$g_{sl3}(\mathbb{x}_{sl}) \equiv ||r(\tau) - r_d||^2 = 0 \tag{22d}$$

$$g_{sl4}(\mathbb{x}_{sl}) \equiv ||v_\tau||^2 = 0 \tag{22e}$$

The simulation was run with the bounds on the design variables as $m^{(b)} = \begin{bmatrix} 1 & 8 \end{bmatrix}$ kg and $r^{(b)} = \begin{bmatrix} 8 & 30 \end{bmatrix}$ cm. Figure 10a shows the average cost of the population and the cost of the best/worst individual in the population. The best individual found after 40 generations is with the design variables, mass $m = 4.35$ kg and radius $r = 16$ cm. Table 4 shows optimal values of the relevant properties of each subsystem for the optimal design of SphereX. Figure 10b shows the 3D CAD model of the optimal design for the exploration of the Mare Tranquilitatis pit.

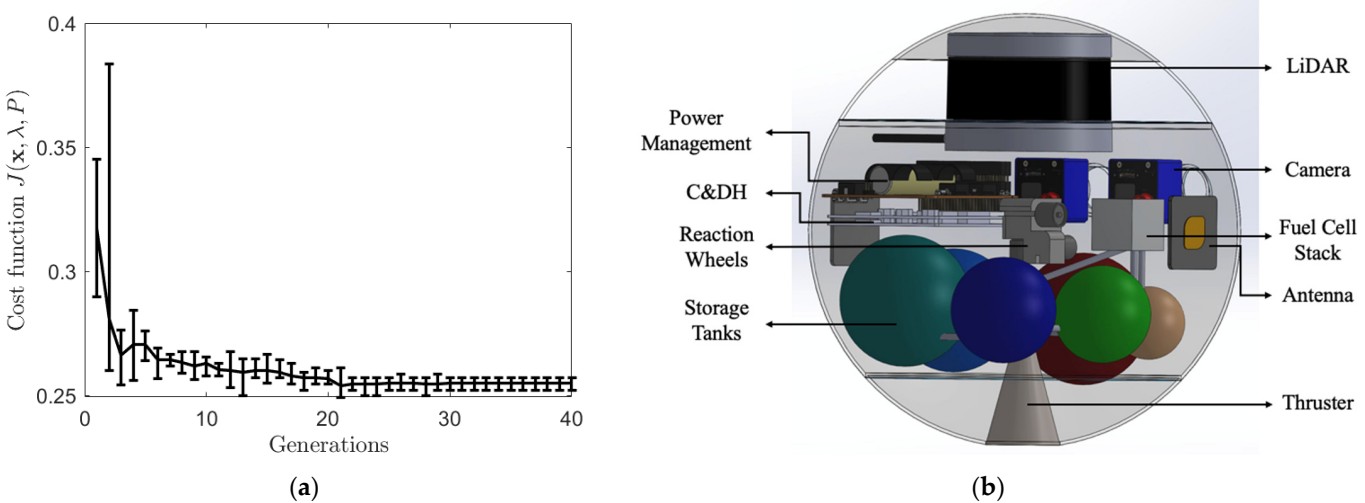

**(a)**　　　　　　　　　　　　　　　　　　　　　　**(b)**

**Figure 10.** (**a**) Average cost along with best and worst cost across number of generations. (**b**) A fully assembled CAD model of the optimal design.

**Table 4.** Optimal values of relevant properties of each subsystem for SphereX.

| Subsystem | Variable | Scenario 1 | Scenario 2 |
|---|---|---|---|
| Fuel Cell | No. of cells, $n$ | 3 | 3 |
|  | Current density, $i$ | 194.38 (mA/cm$^2$) | 194.38 (mA/cm$^2$) |
|  | Voltage of each cell, $V$ | 0.9146 (V) | 0.9146 (V) |
| Propulsion | Combustion pressure, $p_c$ | 3 (MPa) | 3 (MPa) |
|  | Exit pressure, $p_e$ | 103.57 (Pa) | 309.88 (Pa) |
|  | Mixture ratio, $r_m$ | 4.2 | 4.06 |
|  | Contraction area ratio, $\epsilon_c$ | 50.06 | 50.03 |
|  | Nozzle throat radius, $r_{th}$ | 0.826 (mm) | 1.2 (mm) |
|  | Nozzle exit radius, $r_e$ | 30.2 (mm) | 26.5 (mm) |
|  | Nozzle length (diverging), $L_{n(div)}$ | 109.8 (mm) | 94.5 (mm) |
|  | Nozzle length (converging), $L_{n(con)}$ | 4.6 (mm) | 5.0 (mm) |
|  | Combustion chamber radius, $r_c$ | 5.8 (mm) | 6.4 (mm) |
|  | Combustion chamber length, $L_c$ | 24 (mm) | 24 (mm) |

**Table 4.** *Cont.*

| Subsystem | Variable | Scenario 1 | Scenario 2 |
|---|---|---|---|
| Chemicals | Mass of LiH, $m_{LiH}$ | 398.2 (g) | 251.1 (g) |
| | Mass of LiClO$_4$, $m_{LiClO_4}$ | 745.8 (g) | 452.1 (g) |
| | Mass of H$_2$O, $m_{H_2O}$ | 872.9 (g) | 560.7 (g) |
| | Mass of N$_2$, $m_{N_2}$ | 34.5 (g) | 21.0 (g) |
| Storage Tanks | Mass of LiH tank, $m_{LiH\ tank}$ | 62.4 (g) | 45.1 (g) |
| | Mass of LiClO$_4$ tank, $m_{LiClO_4\ tank}$ | 37.8 (g) | 26.5 (g) |
| | Mass of H$_2$O tank, $m_{H_2O\ tank}$ | 57.3 (g) | 41.7 (g) |
| | Mass of N$_2$ tank, $m_{N_2\ tank}$ | 53.0 (g) | 32.2 (g) |
| | Mass of H$_2$ tank, $m_{H_2\ tank}$ | 46.1 (g) | 28.3 (g) |
| | Mass of O$_2$ tank, $m_{O_2\ tank}$ | 12.8 (g) | 7.5 (g) |
| Battery | Capacity, $Q$ | 280 (mAh) | 198 (mAh) |
| | No. of batteries, $n_{bat}$ | 1 | 1 |

### 4.2. Test Scenario 2: Exploration of Victoria Crater

The second simulation was run to perform the exploration of the Victoria crater on the surface of Mars. The Victoria crater located at 2.05° S 5.50° W of the planet Mars is an impact crater and has a diameter of approximately 750 m. The Mars Exploration Rover Opportunity travelled 21 months to the Victoria crater and reached its edge on 26 September 2006, which is named "Duck Bay". However, the rover could not explore the interiors of the crater. The concept of operations for exploring such a crater is shown in Figure 11a. A rover carrying multiple SphereXs approaches the edge of the crater and deploys them one by one. Each SphereX then enters the crater by hopping and reaches the crater base to perform experiments. Figure 11b shows the successive hopping trajectories of SphereX for exploring Victoria crater on Mars, from which the mission target is calculated as to explore 750 m inside the Victoria crater in 3.75 h.

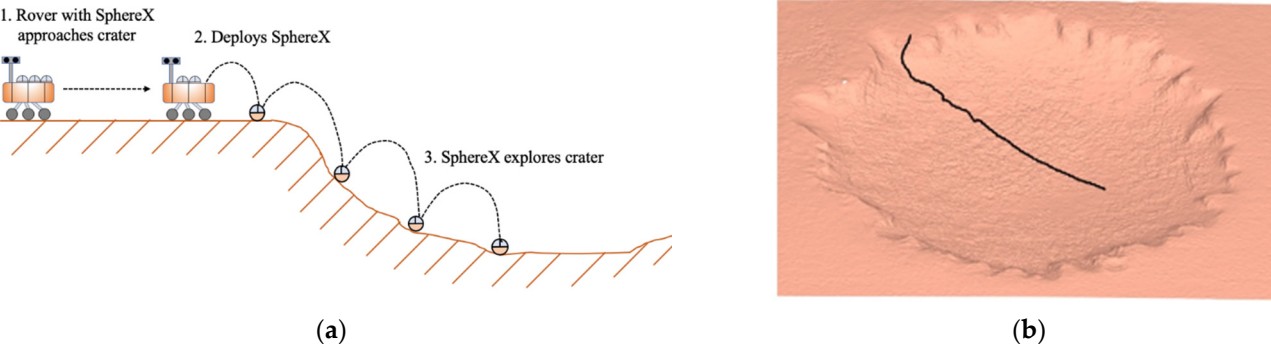

**Figure 11.** (**a**) Concepts of operation for exploring planetary craters. (**b**) Hopping trajectory of SphereX for exploring Victoria crater on Mars.

The simulation was run with bounds on the design variables of $m^{(b)} = \begin{bmatrix} 1 & 8 \end{bmatrix}$ kg, and $r^{(b)} = \begin{bmatrix} 8 & 30 \end{bmatrix}$ cm. The best individual found after 40 generations is with the design variables mass $m = 3.5$ kg and radius $r = 15$ cm. Table 4 shows the optimal values of relevant properties of each subsystem for the optimal design of SphereX.

### 5. Comparative Analysis

The proposed power and propulsion system can power the proposed robotic system SphereX for its mission lifetime $\Gamma$ and explore a target distance of $d_{targ}$ on a target environment using hopping mobility. We compared the proposed combined power and mobility system with other power and mobility systems in terms of its mass for exploring a target exploration distance. For all the hopping mobility systems with propulsion, the single hopping distance is restricted to $d_{hop} = 10$ m for our analysis, and the robotic shell is

designed such that it can maintain its structural rigidity while performing a ballistic hop of 10 m on the selected target environment. The other power systems against which our proposed power system is compared are (a) Direct Methanol Fuel Cell (DMFC), (b) Direct Borohydride Fuel Cell (DBFC), Lithium-Ion batteries, and Lithium-Polymer batteries. Two other hopping systems are compared against the proposed hopping system, which are (a) steam propulsion and (b) mechanical hopping mechanism, the details of which are provided in Appendix D.

*Mass Comparison*

Figure 12 shows the mass budget of SphereX for various combinations of power and the hopping system for the two test scenarios presented in Section 4. The numbers along the *x*-axis represents 1. FC(LiH)-$H_2/O_2$, 2. DMFC- $H_2/O_2$, 3. DBFC- $H_2/O_2$, 4. Li-Ion-$H_2/O_2$, 5. Li-Polymer- $H_2/O_2$, 6. FC(LiH)-Steam, 7. DMFC-Steam, 8. DBFC-Steam, 9. Li-Ion-Steam, 10. Li-Polymer-Steam, 11. FC(LiH)-Mechanical, 12. DMFC-Mechanical, 13. DBFC-Mechanical, 14. Li-Ion-Mechanical, and 15. Li-Polymer-Mechanical. It can be seen that SphereX with LiH fuel cell and $H_2/O_2$ propulsion is the optimal choice for both scenarios.

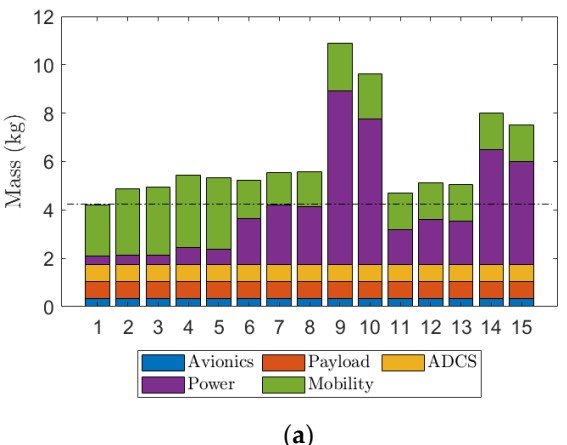

(**a**)

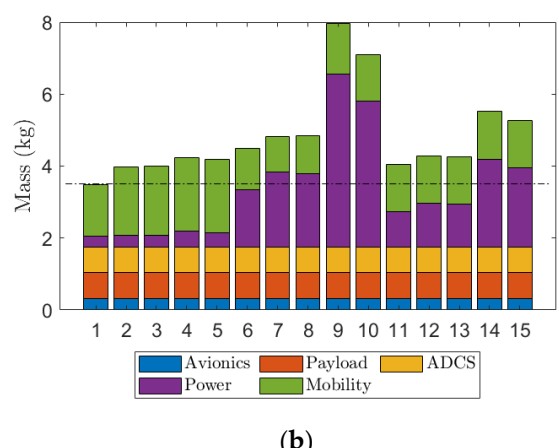

(**b**)

**Figure 12.** Mass budget of SphereX for various combinations of power and hopping system for (**a**) test scenario 1 and (**b**) test scenario 2.

Figure 13 shows the power budget for the three mobility systems. It can be seen that the power requirement for SphereX with $H_2/O_2$ propulsion is the minimum, and that, with steam, propulsion is the maximum. Figure 13b shows that 75% of the power is required to heat the water into steam. As such, the mass of the power system for SphereX with five combinations of power and steam propulsion is the maximum, as seen in Figure 12.

Figures 14 and 15 shows the mass of SphereX for various combinations of power and hopping mobility for varying exploration distances on the surface of the moon and Mars respectively. Figures 14a and 15a show combinations of the five power systems with $H_2/O_2$ propulsion, Figures 14b and 15b show that with steam propulsion, and Figures 14c and 15c show that with mechanical hopping. It can be seen that, for each of the three hopping systems, the combination with the LiH fuel cell has the minimum mass. It can also be seen that, for exploration objectives less than 300 m, the combination with lithium-polymer batteries is better, but, as the exploration objectives increases, the mass of the lithium-polymer batteries increases significantly, making a LiH fuel cell the better option for long-distance exploration. For low-exploration objectives, the masses of SphereX with the three fuel cells (LiH fuel cell, DMFC, DBFC) are approximately the same, but, as the exploration objectives increase, the difference in mass is significant, making a LiH fuel cell the optimal choice. Figures 14d and 15d show the comparison of the mass of SphereX for varying exploration distance with the combination of a LiH fuel cell and the three hopping options.

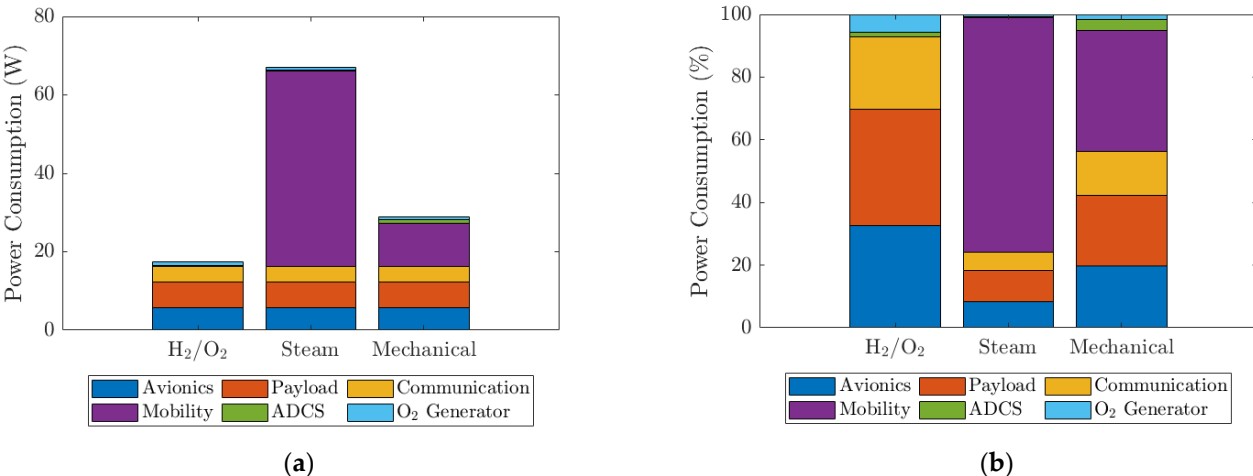

**Figure 13.** Power budget of SphereX for the three hopping systems used for the comparative analysis: (**a**) absolute power budget and (**b**) power budget in percentage.

**Figure 14.** Mass of SphereX for varying exploration distance on the surface of the moon for different combinations of power and propulsion systems: (**a**) for all power systems against $H_2/O_2$ propulsion system; (**b**) for all power systems against steam propulsion system; (**c**) for all power systems against mechanical hopping system; (**d**) for LiH fuel cell power system against all propulsion systems.

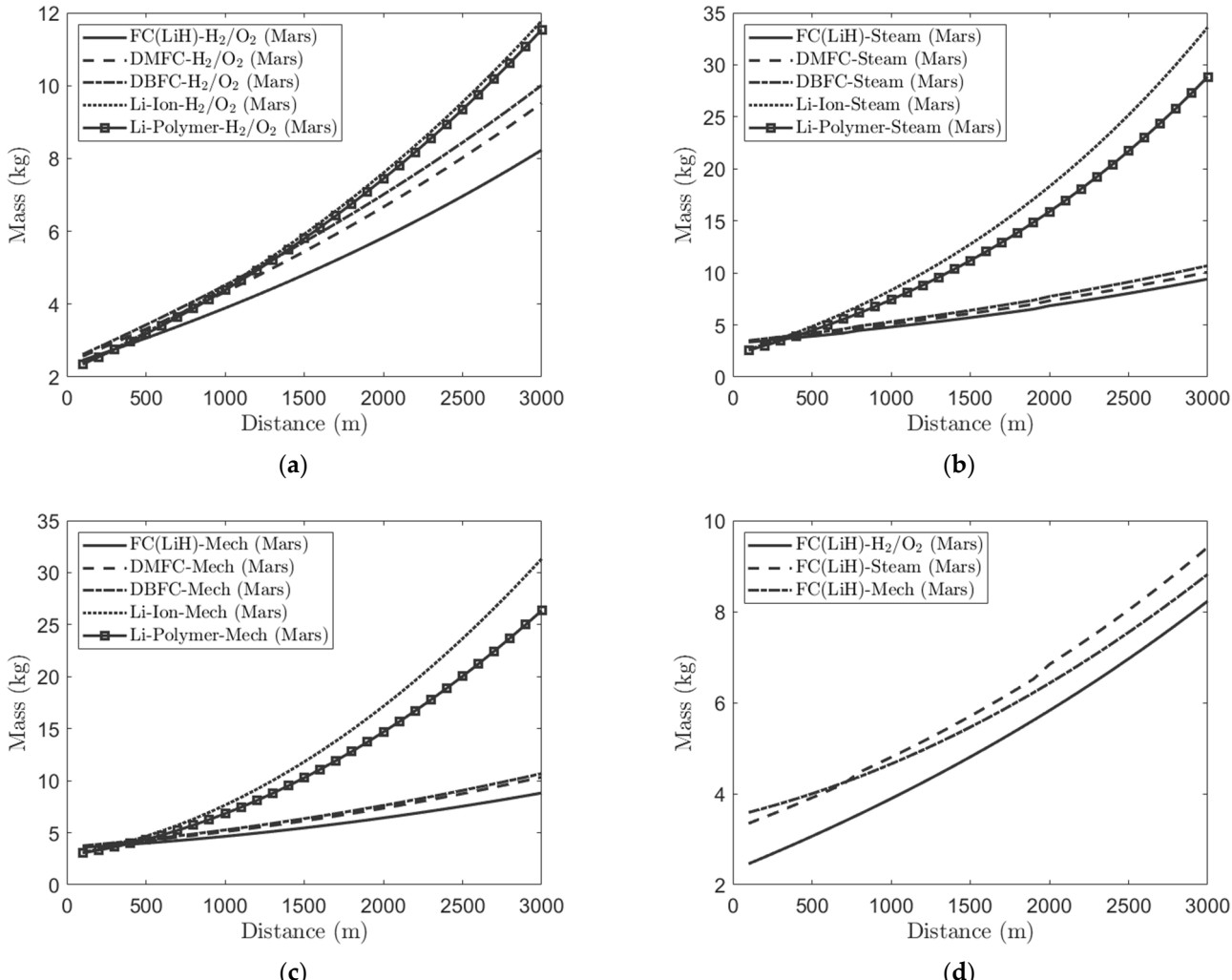

**Figure 15.** Mass of SphereX for varying exploration distance on the surface of Mars for different combinations of power and propulsion systems: (**a**) for all power systems against $H_2/O_2$ propulsion system; (**b**) for all power systems against steam propulsion system; (**c**) for all power systems against mechanical hopping system; (**d**) for LiH fuel cell power system against all propulsion systems.

For this design problem, we compare the performance of Evolutionary Algorithms (EA), Particle Swarm Optimization, and Simulated Annealing [48]. Each of these approaches use stochastic techniques. Figure 16 shows the typical results we obtain for the mean of the fitness function over generations for all the three algorithms simulated and averaged over 20 times. Simulated annealing faces performance challenges and results in the premature convergence of poor solutions. Particle Swarm and Evolutionary Algorithms find feasible solutions, but, overall, we found that EA showed improved computational performance, producing solutions in half the time of PSOs, making it the better choice for this problem. Our observations also show that PSOs stagnate prematurely (more often), while EAs are more effective in continuing to innovate over the generations.

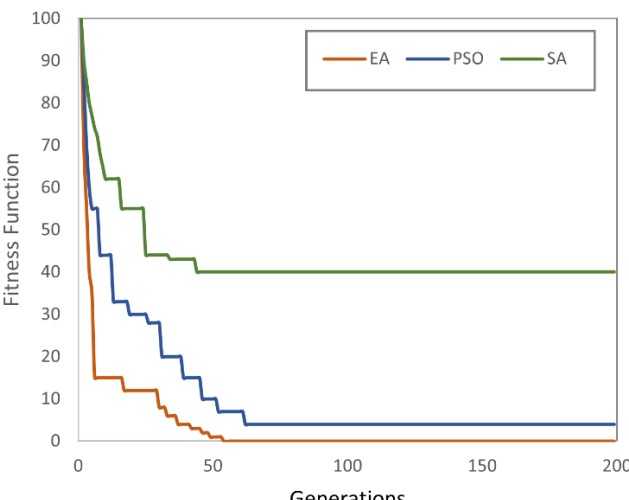

**Figure 16.** Comparison of performance of evolutionary algorithm (EA) against particle swarm optimization (PSO) and simulated annealing (SA) in terms of average fitness over 20 runs.

## 6. Conclusions

We propose to use teams of small hopping robots called SphereXs to explore extreme and rugged terrains on the moon and Mars. Here, we present an integrated power and propulsion system for SphereX containing a PEM fuel cell to generate electricity and a $H_2/O_2$ mobility/propulsion system in which $H_2$ and $O_2$ are stored in solid form. Lithium hydride was selected as the hydrogen source because of its high hydrogen content and the simple hydrolysis reaction required to release the hydrogen. Lithium perchlorate was selected as the oxygen source due to its high oxygen content and relatively simple catalytic decomposition reaction. The power system consists of a fuel cell-battery hybrid configuration to maximize operating life while operating at varying load conditions. The propulsion system, operating along with the attitude-control system, will provide mobility for SphereX in the form of ballistic hopping. We then formulated mathematical models for all the relevant subsystems of SphereX and applied a Multidisciplinary Design Optimization (MDO) methodology to determine an optimal design configuration of SphereX for given mission specs. To solve the MDO task, our approach utilizes a Genetic Algorithm (GA) applied at the system level integrated with multiple gradient-based optimization techniques applied at the subsystem level. This approach finds the near-optimal design configuration in terms of mass, volume, and feasibility of assembly for various missions. Our results present a near-optimal design of SphereX for two test cases: (a) the subsurface exploration of Mare Tranquilitatis on the moon, and (b) the exploration of the Victoria crater on Mars. We presented a comparative analysis of the proposed power and propulsion system with four other power systems and two other hopping systems. The comparative study showed an advantage in using our proposed system for SphereX over other design strategies to minimize the launch mass and volume.

## 7. Patents

The authors Himangshu Kalita and Jekan Thangavelautham report a pending patent on Spherical Robots for Off-World Surface Exploration.

**Author Contributions:** Conceptualization, H.K. and J.T.; methodology, H.K.; software, H.K.; validation, H.K. and A.D.-F.; formal analysis, H.K.; investigation, H.K.; resources, J.T.; writing—original draft preparation, H.K.; writing—review and editing, J.T. and H.K.; visualization, H.K.; supervision, J.T.; project administration, J.T.; funding acquisition, J.T. All authors have read and agreed to the published version of the manuscript.

**Funding:** This research was funded by National Aeronautics and Space Administration, grant number 80NSSC19M0197.

**Institutional Review Board Statement:** Not applicable.

**Informed Consent Statement:** Not applicable.

**Data Availability Statement:** Not applicable.

**Acknowledgments:** We would like to acknowledge Rachel Moses and Troy Jameson for helping with the CAD models.

**Conflicts of Interest:** The authors declare no conflict of interest.

## Appendix A

A PEM fuel cell contains a negatively charged electrode (anode), a positively charged electrode (cathode), and a polymer electrolyte membrane (PEM) sandwiched in between [49,50]. On the anode, hydrogen is oxidized, and, on the cathode, oxygen is reduced. Protons travel from the anode to the electrolyte membrane and into the cathode, while the electrons transported through an external circuit. In a fuel cell, hydrogen protons stay in their ionic state by traveling from molecule to molecule through the polymer membrane. Electrons flow through external circuit(s) to the electrical load(s) when needed. On the cathode oxygen reacts with protons travelling though the membrane and electrons travelling through the external circuit, forming water as a byproduct, and producing heat. To speed up the electrochemical processes, both the anode and cathode are assisted by a catalyst layer. The gas diffusion layers (GDL), catalyst layers and the polymer membrane are sandwiched together to form the membrane electrode assembly (MEA) as shown in Figure A1.

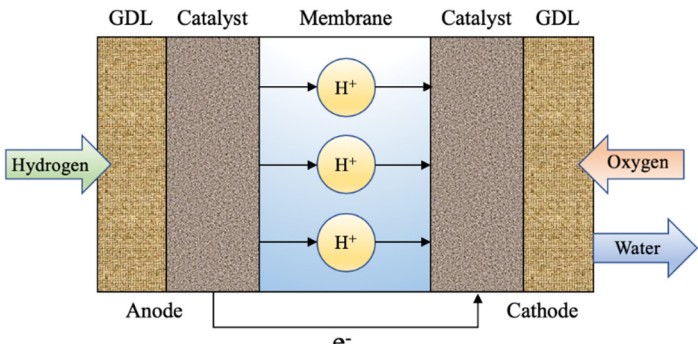

**Figure A1.** A Polymer Electrolyte Membrane (PEM) fuel cell and its major components.

A typical PEM fuel cell has the following reactions:

Anode: $H_2$ (g) → 2H+ (aq) + 2e-; Cathode: $\frac{1}{2}$ $O_2$ (g) + 2H+ (aq) + 2e- → $H_2O$ (l)

Overall: $H_2$ (g) + $\frac{1}{2}$ $O_2$ (g) → $H_2O$ (l) + electrical energy + waste heat

From the reaction equations of the fuel cell, it is evident that four electrons are transferred for each mole of oxygen. So, the rate of oxygen used by the fuel cell is given by $\dot{O}_2 = I/4F$ moles/s. Where, $I$ is the current of a single cell and $F$ is the Faraday's constant. For a stack of $n$ cells, $\dot{O}_2 = In/4F$ moles/s. If the voltage of each cell in the stack is $V$, then power, $P = VIn$. So, the rate of oxygen used in terms of power can be expressed as $\dot{O}_2 = P/4FV$ moles/s. The molar mass of oxygen is $32 \times 10^{-3}$ kg/mole, so the rate of oxygen used is:

$$\dot{O}_2 = 8.29 \times 10^{-8} \frac{P}{V} \frac{\text{kg}}{\text{s}} \tag{A1}$$

The rate of use of hydrogen is derived in a similar way to oxygen. For each mole of hydrogen two electrons are transferred and the rate of hydrogen used can be expressed as $\dot{H}_2 = P/2FV$ moles/s. Hydrogen's molar mass is $2.02 \times 10^{-3}$ kg/mole. Moreover, for every two electrons one mole of water is produced and the molar mass of water is

$18.02 \times 10^{-3}$ kg/mole. Thus, the rate of hydrogen used, and water produced is calculated as:

$$\dot{H}_2 = 1.05 \times 10^{-8} \frac{P}{V} \frac{\text{kg}}{\text{s}}; \; \dot{H_2O} = 9.34 \times 10^{-8} \frac{P}{V} \frac{\text{kg}}{\text{s}} \tag{A2}$$

The theoretical value of the open circuit voltage (OCV) of a hydrogen fuel cell is given as $E_{oc} = -\Delta \bar{g}_f / 2F$, where $\Delta \bar{g}_f$ is the Gibbs free energy of the basic reaction for the hydrogen/oxygen fuel cell. However, in practice the operating voltage is less than the theoretical value due to losses in the form of ohmic losses, activation losses, and mass transport or concentration losses [33]. Adding all these losses, for a current density $i$ the operating voltage of a fuel cell is expressed as, $V = E_{oc} - ir - A\ln(i) + me^{(ni)}$, where, the open circuit voltage is expressed as $E_{oc}$, the area-specific resistance is expressed as $r$, the slope of the Tafel line for the fuel cell is expressed as $A$, and the mass-transfer loss constants are expressed as $m$ and $n$ as shown in Table A1. Figure A2 shows the variation of cell voltage as a function of current density with the losses included.

**Table A1.** Example constants for voltage.

| Constant | PEM Fuel Cell at 70 °C |
|---|---|
| $E_{oc}$ (V) | 1.031 |
| $r$ (k$\Omega$ cm$^2$) | $2.45 \times 10^{-4}$ |
| $A$ (V) | 0.03 |
| $m$ (V) | $2.11 \times 10^{-5}$ |
| $n$ (cm$^2$ mA$^{-1}$) | $8 \times 10^{-3}$ |

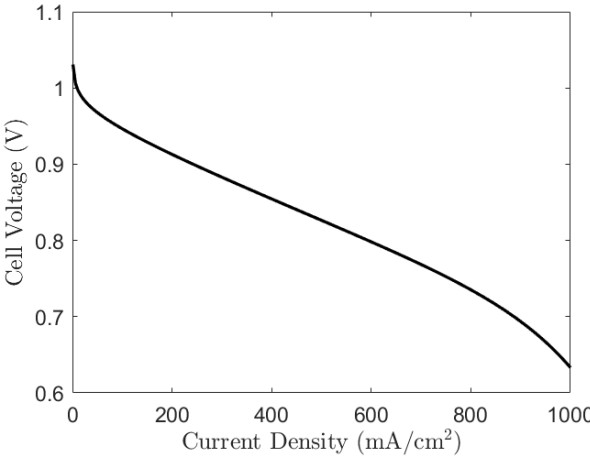

**Figure A2.** Graph of cell voltage against current density with losses.

**Appendix B**

This section presents a brief analysis of the dynamics of the robot while using the combined action of the propulsion system and the attitude-control system for performing ballistic hops. During ballistic hopping, the robot needs to reach a desired state $r_d$ from its initial state $r_0$. SphereX will perform ballistic hopping by applying forces from rest on the surface with no control of the trajectory midflight. Three reaction wheels will be used to adjust its attitude, and one thruster will be used to provide constant thrust along the $b_3$ axis of the body's fixed frame ($\mathcal{B}$), as shown in Figure A3a. The hopping mode is divided into three phases: (a) attitude-correction phase, (b) boost phase, and (c) ballistic trajectory phase. Figure A3c shows the time diagram for the three phases [51]. During the attitude-correction phase, the robot needs to orient itself from its initial attitude state $q_0$ to the desired attitude state $q_d$ in the presence of external disturbances while on ground. For the robot to hop from its initial position $r_0$ to its final position $r_d$, the desired angles are defined as $q_d = [\phi, \pi/2 - \theta, 0]$ as shown in Figure A3b. During this phase, the robot interacts with the surface of the target environment, as such, a surface interaction model

that governs the motion of the rigid sphere body on a deformable terrain (Lunar and Martian soil) is developed [52]. A sliding-mode controller is designed for the robot to attain its desired attitude states during the attitude-correction phase and maintain it during the boost phase, in the presence of external disturbances [51]. Next, during the boost phase, the thruster provides a constant thrust $||F||$ for a burn time $t_b$ with the attitude-control system maintaining the desired attitude states $q_d$. For a single pinpoint hopping movement, to find the optimal burn time $t_b$, the details of the optimal control problem are presented here. The equations governing the motion of the robot can be expressed as:

$$\dot{r} = v, \quad \dot{v} = \begin{cases} g + \frac{F}{m} \\ g \end{cases}, \quad \dot{m} = \begin{cases} -\frac{|F||}{I_{sp}g_0} & if \ t < t_b \\ 0 & if \ t_b < t < \tau \end{cases} \tag{A3}$$

The key objective of the optimization step is to find $\mathbb{x}_h = [t_b, \tau]$ for the optimal values of the design variables, such that the fuel used for hopping is minimized. Two constraints are added, such that the burn time $t_b < \tau$ and the final position of the robot is equal to its desired position as $||r(\tau) - r_f||^2 = 0$. The optimization model can be mathematically expressed as:

$$\min f(\mathbb{x}_h) = \int_0^{t_b} ||F||dt \tag{A4a}$$

$$subject \ to: \ g_1(\mathbb{x}_h) \equiv t_b - \tau < 0 \tag{A4b}$$

$$g_2(\mathbb{x}_h) \equiv ||r(\tau) - r_d||^2 = 0 \tag{A4c}$$

Finally, during the ballistic trajectory phase, there is no control over the robot, and it follows the trajectory based on its dynamics.

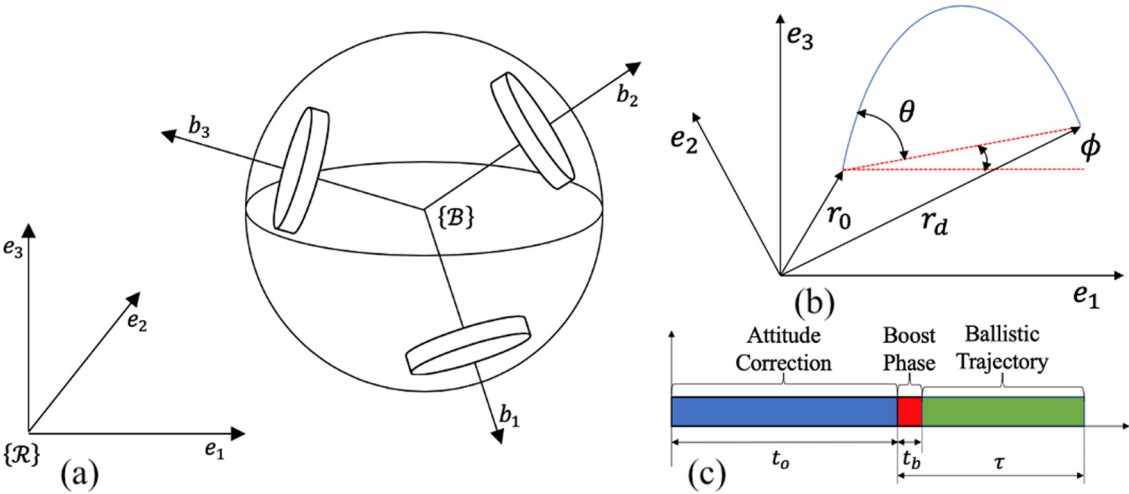

**Figure A3.** (**a**) Reference frame $\{\mathcal{R}\}$ and body fixed frame $\{\mathcal{B}\}$; (**b**) Position vectors and angles for ballistic hopping; (**c**) Time diagram for ballistic hopping.

## Appendix C

This section provides the details of the attitude control, command and data handling, communication, and instruments subsystems used to design SphereX.

### Appendix C.1. Attitude Control

SphereX will use three-axis reaction wheel system for maintaining roll, pitch, and yaw angles and angular velocities along its principal axes for executing ballistic hopping. For simplicity, we have considered commercially-off-the-shelf reaction wheels for this paper. The selected reaction wheel is RWP050 from Blue Canyon Technology which has a mass of 0.24 kg and dimensions $58 \times 58 \times 25$ mm. Each wheel has a momentum of 0.050 Nms and a maximum torque of 7 mNm and consumes < 1 W power operating at full momentum.

An MAI400 Controller Board is selected to control the reaction wheels that integrate the four microcontrollers with three sets of reaction wheel drive circuitry, a three-axis MEMS accelerometer, and a three-axis MEMS gyroscope. The board consumes only 0.45 W power under nominal conditions, weighs only 47 g, and has a dimension of 86 × 88 mm.

*Appendix C.2. Command and Data Handling*

The main computer selected is Rincon Research's AstroSDR which an ARM processor, FPGA signal processor, data-storage capabilities, and a software-defined radio (SDR). The ARM Cortex A9 with NEON processor can operate at up to 733 MHz. For storage, it contains 512 Mbyte DDR3 RAM memory, 2 GByte Flash, and an optional 64 GByte eMMC flash storage. Receivers and transmitters connected to the SDR for communications can be tuned between 70 MHz to 6 GHz with a max bandwidth of 56 MHz. The board consumes 5.5 W power under nominal conditions, weighs only 95 g, and has a dimension of 90 × 90 mm.

*Appendix C.3. Communication*

With multiple SphereX deployed to cooperatively explore through a target area, efficient communication is a key factor. The main computer selected (AstroSDR) consists of a software-defined radio (SDR), with a tuning range of 70 MHz to 6 GHz, which will be used as the transceiver. For antennas, we will use two S-band antennas. The selected antenna is the S-band patch antenna from EnduroSat, which can operate in the 2.4–2.45 GHz bandwidth. Each antenna consumes power up to 4 W, has a weight of 64 g, and dimensions 98 × 98 × 5.5 mm.

*Appendix C.4. Instruments*

SphereX will carry a pair of stereo cameras for imaging and a 3D LiDAR for navigation and mapping. The robot will use an Iterative Closest Point (ICP)-based Pose-Graph Simultaneous Localization and Mapping (SLAM) algorithm on scans generated by the LiDAR for both navigation and mapping [51]. The camera selected is the mvBlueFOX3 by Matrix Vision, which has a mass of 58.5 g and dimensions 39 × 39 × 29 mm. The sensor has an image resolution of 1280 × 960 and consumes ≤ 2.5 W power. The LiDAR selected is Velodyne LiDAR's Puck LITE, which has a mass of 0.59 kg, dimensions of height 71.7 mm, and diameter 103.3 mm and consumes 8 W power.

Table A2 shows the I/O interfaces of each component selected for SphereX which shows their compatibility.

**Table A2.** Available I/O interfaces for each component selected for SphereX.

| Component | Available Interfaces |
| --- | --- |
| AstroSDR | GPIO, USB, UART, LVDS, Ethernet |
| MAI400 | UART, RS232, I2C, SPI |
| RWP050 | I2C |
| S-band patch antenna | UART |
| mvBlueFOX3 | USB |
| Puck LITE | Ethernet |
| NanoPower P31u board | I2C |

**Appendix D**

This section provides a method to assemble all the components of SphereX inside the robot shell. A program to assemble all the components automatically is developed. Separate sequences are written for each component that contain their physical dimensions. For the final assembly, first we create a sphere of radius $r$ and then assemble the propulsion subsystem which occupies the lower half of SphereX. The nozzle, combustion chamber, catalyst bed, and injector are stacked vertically and placed such that the center of the nozzle

exit is vertically below the center of the sphere and aligned with the bottom surface of the sphere. Next, the hydrogen generator, oxygen generator, water, oxygen, hydrogen, and nitrogen tanks are assembled around the nozzle assembly. A binary assembly index $Index_p$ is provided for the propulsion system based on its feasibility for assembly. Next, on top of the propulsion system, we assemble the attitude-control system, avionics, batteries, fuel cell stack, and cameras. The attitude-control system is assembled along the center of the sphere, and the avionics boards, the batteries, the fuel cell stack, and the cameras are assembled around the attitude-control system. Another binary assembly index $Index_a$ is provided for the ADCS, avionics, battery stack, and cameras based on its feasibility for assembly. On top of this, the LiDAR sensor is assembled that provides another binary assembly index $Index_l$ based on its feasibility of assembly. All the components are separated by a distance $\varepsilon$. The program checks for the intersection of each component with the robot shell and returns a value of 1 if there is no intersection and a value of 0 otherwise as shown in Figure A4. Finally, an AND logical operator to determine the assembly index of the entire system as $Index = Index_p$ & $Index_a$ & $Index_l$, which is used as a constraint for the system-level optimization problem.

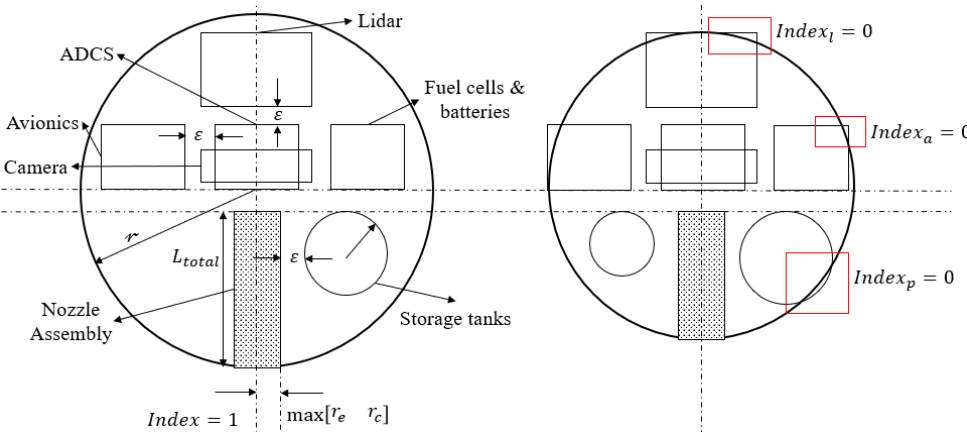

**Figure A4.** Schematic of automated assembly of the components for SphereX. The red squares shows when any component intersects with the shell, the program returns an index of value 0.

**Appendix E**

This section provides the details of the alternate power systems and hopping systems used for the comparative analysis in Section 5.

*Appendix E.1. Alternative Power Systems*

The other power systems against which our proposed power system is compared are (a) Direct Methanol Fuel Cell (DMFC) and (b) Direct Borohydride Fuel Cell (DBFC), with sodium borohydride, Lithium-Ion batteries, and Lithium-Polymer batteries. The details of these alternative power systems are discussed below.

Appendix E.1.1. Direct Methanol Fuel Cells (DMFC)

The DMFC is a proton-exchange membrane (PEM) fuel cell that uses methanol as a fuel which oxidizes to produce water and carbon dioxide. At the anode, water reacts with methanol to produce protons. The protons are transported across the polymer membrane and react with oxygen at the cathode to produce water. Electrons are transported through an external circuit when connected to a load from the anode to the cathode. The half-reactions for the DMFC are:

Anode: $CH_3OH + H_2O \rightarrow 6H+ + 6e- + CO_2$; Cathode: $3/2\, O_2 + 6H+ + 6e- \rightarrow 3H_2O$

Overall: $CH_3OH + 3/2\, O_2 \rightarrow 2H_2O + CO_2$ + electrical energy + waste heat

Appendix E.1.2. Direct Borohydride Fuel Cells (DBFC)

The DBFC is a PEM fuel cell, that uses sodium borohydride as a fuel to produce water and sodium meta-borite. The half-reactions for the DBFC are:

Anode: $NaBH_4 + 2H_2O \rightarrow 8H+ + 8e- + NaBO_2$; Cathode: $2O_2 + 8H+ + 8e- \rightarrow 4H_2O$

Overall: $NaBH_4 + 2O_2 \rightarrow 2H_2O + NaBO_2$ + electrical energy + waste heat

For both the DMFC and DBFC, oxygen is generated on demand by the decomposition of LiClO4 as shown in Section 3.4.

Appendix E.1.3. Lithium-Ion and Lithium-Polymer Batteries

The power required for the entire mission can be stored through lithium-ion or lithium-polymer batteries that have a specific energy of 100–265 Wh/kg and an energy density of 250–693 Wh/L and 250–730 Wh/L, respectively. For our analysis, we have selected the GomSpace NanoPower Lithium Ion 18650 battery that has a capacity of 2600 mAh @ 3.7 V, mass of 48 g, and dimensions of length 65 mm and radius 18.5 mm, and the Pegasus Class BA0x lithium polymer battery, where each cell has a capacity of 1500 mAh @ 3.7 V, mass of 27 g, and dimensions of $85 \times 22 \times 7$ mm. The State of Charge (SOC) of the battery can be computed by using (24–26) with $P_{fc} = 0$, as discussed in Section 3.6.1. With the model defined, we calculate the number of batteries required to supply a power $P$ for time $\Gamma$, which provides the mass and volume of the respective battery power systems.

*Appendix E.2. Alternative Hopping Systems*

Two other hopping systems are compared against the proposed hopping system. The propulsion system required to provide the required thrust can be conducted through steam propulsion, or a mechanical hopping mechanism can be used to provide the required thrust. The details of those two systems are provided below.

Appendix E.2.1. Steam Propulsion

For steam propulsion, water is heated to high-temperature steam through electrical heating and then used for steam propulsion. The ambient pressure and temperature on the Lunar and Martian surfaces are below the triple point of water, and, as such, water is stable only in solid and vapor states. The water in solid/vapor states is heated to a temperature $T_c$ from its ambient temperature $T_a$ through electrical heating and then pressurized to a pressure of $p_c$ using pressurized nitrogen gas before being fed into the nozzle. The energy consumed $E_h$ for heating water from $T_a$ to $T_c$ is calculated as follows:

$$E_h = m_{prop}\left(c_{p,s}(T_s - T_a) + \Delta H_s + c_{p,v}(T_c - T_s)\right), \tag{A5}$$

where, $c_{p,s}$ and $c_{p,v}$ are the specific heat of solid and vapor, $\Delta H_s$ is the heat of sublimation, and $T_s$ is the sublimation temperature. Thus, the power required for heating water is calculated as $P_h = E_h/\Gamma$, where $\Gamma$ is the mission lifetime. Figure A5 shows the variation of $I_{sp}$ of steam and energy required per kg of steam to heat from $T_a = -25$ °C as a function of $T_c$ at a constant pressure $p_c = 2$ MPa. As $T_c$ increases, the specific impulse increases, so does the energy required to heat it from the ambient temperature. Hence, finding the optimal value of $T_c$ is important to making a valid comparison.

The objective of the optimization process for the steam propulsion is to minimize the total mass of the propulsion system $m_{p(total)} = m_{dry} + m_{prop}$ and the mass of the power system required to heat the water into steam $m_{power}$. The design variables are combustion pressure $p_c$, exit pressure $p_e$, contraction area ratio $\epsilon_c$, and combustion temperature $T_c$, $\mathbb{x}_p = [p_c, p_e, \epsilon_c, T_c]$. One constraint is added to the optimization function, such that the total length $L_{total}$ is less than 90% of the radius of the robot. As such, the input the mass $m$

and radius $r$ of the robot, the target exploration distance $d_{targ}$, and single hopping distance $d_{hop}$ should be given. The optimization problem is described here:

$$\min f_p(\mathbb{x}_p) = m_{p(total)} + m_{power} \tag{A6a}$$

$$subject\ to: \ g_p(\mathbb{x}_p) \equiv L_{total} < 0.9r \tag{A6b}$$

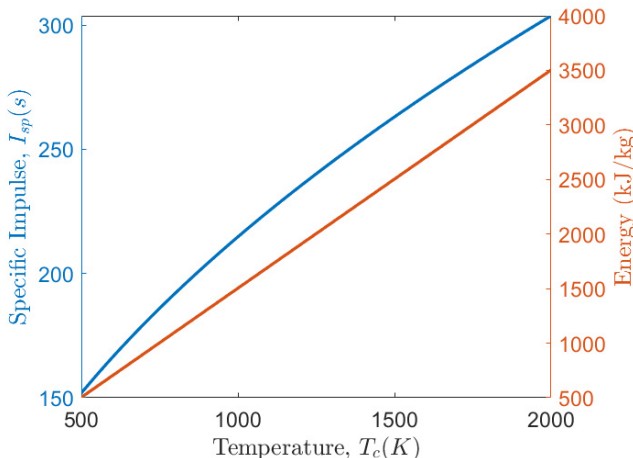

**Figure A5.** Variation of $I_{sp}$ of steam and energy required per kg of steam to heat from its ambient temperature as a function of temperature.

Appendix E.2.2. Mechanical Hopping

The second option for comparison is a spring-and-gear-based mechanical hopping mechanism. The mechanism consists of two mechanical subsystems, one for producing a reactive thrust and the other to adjust the robot attitude into a desired orientation. The combined action of the two mechanical subsystems allows the robot to perform ballistic hopping as shown in Figure A6. The robot's interior consists of a geared motor connected to a rack and a foot through a spring that, when compressed and then released, produces a reactive thrust along the longitudinal axis of the robot. To orient the robot, the other mechanical subsystem consists of three linear actuators, each connected to levers mounted to the robot's shell. Actuating the levers lets the robot achieve its desired orientation, and then the hopping mechanism is initiated to launch the robot in a desired ballistic trajectory. Our past work has presented an optimization technique to find the optimal mass and size of this hopping mechanism to perform hopping on planetary bodies with varying gravity, and which we use here [53].

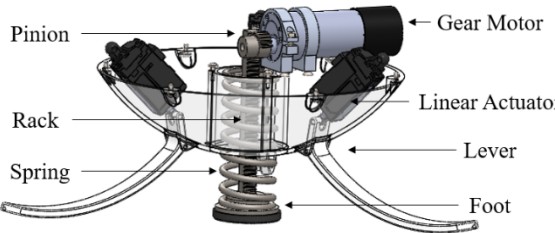

**Figure A6.** 3D CAD model of the mechanical hopping mechanism.

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
