# Peer review of "Integrated Power and Propulsion System Optimization for a Planetary-Hopping Robot"

_aerospace, doi:10.3390/aerospace9080457_

Round 1

Reviewer 1 Report

This paper presents an integrated power and propulsion system optimization for a planetary hopping robot. Basically, I think this paper needs minor revision to meet the requirements of the Aerospace journal. Some suggestions are listed below for possible further improvement of this paper.

  1. It is suggested to increase the comparison between the optimization design method proposed in this paper and other methods in terms of mass, volume and power.
  2. It is suggested to add ground prototype for verification.

Author Response

Please see our response in the attached file.

Reviewer 2 Report

Dear authors, thank you for your interesting paper.

In order to improve I have a few suggestions:

1) In general the paper is too long, you should try to cut parts that describe known technologies, e.g. chapter 3.1 about PEM fuel cell

2) you should explain the origin of expression (18) that you use to size the shell  

3) you should comment the proposal of using carbon fiber for the shell, as it is going to undergo repeated shocks 

4) please review figure 18 and 19: they could be in color, to identify the lines on the same plot, or in black and white (better in color). Certainly not in red and white as fig. 19 is

5) in the final part of the paper you report a detailed CFD and FEA analysis of the thrusters. This part of the paper is not consistent with the former one, which is about the design of th sphereX robot. You should wonder if you really want to include it in the paper. In the case, you should explain which information you obtain in order to describe the system design, possibly by comparing the results that you obtain by CFD with the assumptions that you made to size the thrusters

In general the paper wold be easier to read if it was 20 - 25 pages long

Author Response

Please find our response in the document attached.

Thanks

Round 2

Reviewer 2 Report

The authors accepted all suggestions and answered all the point that I posed in my first review. The paper is now ready for publication

This manuscript is a resubmission of an earlier submission. The following is a list of the peer review reports and author responses from that submission.

Round 1

Reviewer 1 Report

The respected authors present a rolling/hopping robot system dubbed SphereX for the exploration of extreme environments on celestial bodies. The manuscript describes in detail different aspects of the overall system architecture, such as energy generation via PEM fuel cells, hydrogen and oxygen generators, as well as attitude controller, propulsion system, storage tanks, as well as other system-critical components, are introduced and design implementations discussed. Genetic Algorithms were applied to optimize all the above systems and their key system parameters for a specific mission environment and mission scenario. Two mission scenarios are explained in detail and also presented in a comparative analysis.

The paper is well written, attractive to a broad range of readers, topical, and comprehensive. Sometimes, the manuscript is rather lengthy and could profit from moving certain detailed discussions, calculations, and figures to a Supplementary Materials document. However, in my humble opinion, 28 pages are too long, and a shorter manuscript could potentially attract more readers.

Please find below my inputs/concerns:

Minor:

- while it is an excellent name (SphereX), consider renaming it since there is already a SPHEREx (The Spectro-Photometer for the History of the Universe, Epoch of Reionization and Ices Explorer) mission, led by the California Institute of Technology and NASA JPL.

- The introduction could be a little longer, for example, by adding recent developments enabled e.g. by the Mars helicopter Ingenuity.

- Figures in general: Figure numbers are all over the place (Fig. 1, 2, 1, 2, 5, 3, 7, 4, 9, 5,… etc). Make sure to bring them in order and cite the correct and corresponding figures in the text.

- Page 2, line 58: the acronym PEM has not been introduced yet. (Introduced page 3, line 123)

- Page 2, line 59-61: Generation of H2 and O2: please add references.

- Equation 12: Capital Gamma has previously been introduced as Vandenkerckhove function. I doubt it has the same meaning here. Please change letter or re-introduce it.

- Page 11, line 414. What kind of micropumps were you considering? Please add references, as this is not straightforward.

- 10. Automated assembly: consider adding figures explaining the steps.

- Page 16, line 591: there is a lonely «At» at the end of the sentence.

Major:

- 7. Oxygen Generator: please add references. But more importantly: please describe and discuss in more detail how the required 400 C will be reached and maintained and how much energy will be required for this. Probably quite substantial but cannot be seen in the estimated power consumption profile.

- the previous question rolls into this one: how can it be that in Fig. 12, there is essentially no power required for Mobility? (e.g. heating to 400 C will require quite some power).

Thank you.

Author Response

Please see detailed response attached as pdf.

Reviewer 2 Report

The paper presents the optimization by genetic algorithm of a hopping rover for the exploration of lava tubes. The approach presented is very interesting. The paper is well written although there are some points that should be clarified before final publication.

1) In paragraph 3.1 the optimization of PEM fuel cells, it seems to me that an important parameter is the MEA surface of each cell, this affects the final weight why is it not considered in the cost function?

2) Isn't it convenient to turn the cost function so that the number of cells does not appear, but rather their mass? In the end what you want to optimize is the mass .....

3) Paragraph 5, the requested thrust is equal to F = 2mg, how this paramter has been choosen? Could you justify this value. I think it would be better to start from the maximum distance that you want to reach, then define the speed of detachment (which will change depending on the celestial body considered), and finnaly define the request force.

4) How was the ADCS chosen?Has it been verified that the momentum provided and the torques are sufficient to control the SphereX set-up?

5) In paragraph 9.4. Instruments are presented with sensors for mapping and navigation, but these are useless if they are not associated with specific algorithms. The authors could define which algorithms and navigation techniques for the reconstruction of the trajectory they intend to use, and refer to some work in the literature for the navigation of tumbling rovers.

6) I see the proposed system uses of many systems from different manufacturers: Attitude control form Blue Canyon Technologies, Antenna from Endurosat, Data handling form Rincon Research's AstroSDR and so on .... are you are sure that all these elements communicate with each other despite different communication protocols and firmware? Please verify at least the input and output ports...

Author Response

(The authors gave the same response as above.)

Reviewer 3 Report

The authors present a detailed and extensive description of a multi-agent robotic exploration system built around a mother lander platform and doughter spherical exploration robots named SphereX. The paper in general shows an almost meticulous analysis of most aspects of the SpereX, including the dynamics, the propulsion system, the power production system and the propellant production system; the authors go further by proposing a test-case scenario of the exploration of a crater with real-world elevation maps, and discuss several comparative examples. From a first, partial review of the paper I took a few notes of problems to be addressed, as follows:

- Several typos and wonky syntax in the abstract and introduction.

- A conclusive section seems to be entirely missing.

- Figures are not in order and figure references are wrong in many places, e.g. Fig. 2 is referenced at line 232 as Fig. 4 (I presume). There are 2 Figure 1, 2 Figure 7, 2 Figure 9. 

This last point makes it impossible for me to continue the review of the paper in its current form.

Additionally, it is my opinion that the paper, while very detailed and accurate in its contents, fails in making a well defined point. I suggest the authors decide what is the scope of the paper and remove any non-necessary topics, e.g. the Dynamics of the hopping/landing motion, OR a system-engineering study OR a study on the fuel cell-based power production system OR the propulsion system. In principle the paper could well be split into a series of different papers, which would make it much more readable. The current shape the manuscript is in is more like a design report or white paper than a scientific paper. This is not adequate.

All of this is very unfortunate because I can see that the authors have a wealth of material to work with and to disseminate within the scientific community, all of which I believe is very interesting. I strongly encourage them to resubmit a new paper (or series) with a much more focused approach on selected aspects of the system. I would be exceedingly happy to review the new paper then.

In this form the paper is unpublishable, even with major revisions.

Author Response

(The authors gave the same response as above.)

Round 2

Reviewer 2 Report

The author adress the reviewers concerns and the can be pubblished in its corrente form.

Author Response

Thank you.  Based on the review the papers has met the requirements of the review.

Reviewer 3 Report

The authors did some modifications to the paper, mainly by moving much of the text in appendices. This is ok, in principle, but the paper still lacks proper organization. The positive side is that now there seems to be a focus in the narrative, however, I believe that a lot more effort is required to make the text clear and the research appropriately described. Please consider the following:

In general, the number of sections is much too high, which means that the work is not presented organically following the classical structure: introduction, materials & methods, results, discussion, conclusion. This makes everything very complex to follow and ultimately exceedingly dispersive.

I encourage the authors to reduce the number of sections to at least half of the current number, with a clear distinction between them; e.g. Sections 3-9 should be contained in a single "system design" section, which would stand as the "methodological" part of the paper; indeed, it would contain the optimization section, which is definitely a methodological part. Within this section, the authors can and shall use subsections to properly divide the corpus of the article. Each section shall have a brief explanatory introduction before its subsections are introduced.

The authors should add a short bullet list with the main contributions of their work, in the introduction. Instead, in lines 48-90, the authors describe their approach in a very detailed way, which should in general be avoided in the introduction. This is confusing and premature, and should be moved for example in the introductory section for the methodological section. Also, in the tail of the introduction it is customary (and for a reason!) to describe the layout of the main sections of the paper, by describing their contents.

All of the above is scientific paper writing 101.

Finally, how do the authors propose to validate their approach? It is generally required to perform either an experimental campaign or at the very list to run a simulation using any well known commercial software.

To summarize, while the merit of the work I certainly recognize, I cannot recommend that the paper be accepted in the current form. Its structure is still not sufficiently properly organized; additionally, there is a general lack of validation of the proposed methodology; both of these aspects make the article still very immature.

Author Response

Please see latest response to reviewer 3.  We have tried our best to address all of the latest comments from the review and include a validation section.

Thank You,

Jekan

Round 3

Reviewer 3 Report

To me the article seems still much more a report than a scientific paper.

Previously I recommended the editor to reject the paper; I then proposed the authors to perform a full rewrite, both because the presentation structure was not adequate for a scientific journal and due to a general lack of focus.

At this point, after repeated attempts and nudges, I have to say that in my opinion, the authors did not do a convincing work.

The previously mentioned problems still stand: poor organization, exceedingly long paper, poor readability, lack of scientific focus.

If I may, I'd like to give the authors a suggestion: keep papers short. A world-class paper can fit comfortably in 6 pages, the authors propose 34! Your work can be easily fit into a 10 pages manuscript. Everything else is purely distracting.

It pains me because I can see that there is a lot of work "behind the scenes", but at this point I have to reject the paper.